# Novel features of centriole polarity and cartwheel stacking revealed by cryo-tomography

Sergey Nazarov[1,2,†] (ID), Alexandra Bezler[1,†] (ID), Georgios N Hatzopoulos[1,†] (ID),
Veronika Nemčíková Villímová[1], Davide Demurtas[2], Maeva Le Guennec[3] (ID), Paul Guichard[3] (ID) &
Pierre Gönczy[1] (ID)

## Abstract

Centrioles are polarized microtubule-based organelles that seed the formation of cilia, and which assemble from a cartwheel containing stacked ring oligomers of SAS-6 proteins. A cryo-tomography map of centrioles from the termite flagellate *Trichonympha* spp. was obtained previously, but higher resolution analysis is likely to reveal novel features. Using sub-tomogram averaging (STA) in *T.* spp. and *Trichonympha agilis*, we delineate the architecture of centriolar microtubules, pinhead, and A-C linker. Moreover, we report ~25 Å resolution maps of the central cartwheel, revealing notably polarized cartwheel inner densities (CID). Furthermore, STA of centrioles from the distant flagellate *Teranympha mirabilis* uncovers similar cartwheel architecture and a distinct filamentous CID. Fitting the CrSAS-6 crystal structure into the flagellate maps and analyzing cartwheels generated *in vitro* indicate that SAS-6 rings can directly stack onto one another in two alternating configurations: with a slight rotational offset and in register. Overall, improved STA maps in three flagellates enabled us to unravel novel architectural features, including of centriole polarity and cartwheel stacking, thus setting the stage for an accelerated elucidation of underlying assembly mechanisms.

**Keywords** cartwheel; centriole; microtubules; SAS-6; *Teranympha*; *Trichonympha*
**Subject Categories** Cell Adhesion, Polarity & Cytoskeleton; Structural Biology
**The EMBO Journal (2020) 39: e106249**

## Introduction

Centrioles are evolutionarily conserved microtubule-based organelles that seed the formation of primary cilia, as well as of motile cilia and flagella. Despite significant progress in recent years, the mechanisms orchestrating centriole assembly remain incompletely understood, in part because the detailed architecture of the organelle has not been fully unraveled.

The centriole is a 9-fold radially symmetric cylindrical organelle typically ~500 nm in length and ~250 nm in diameter, which is polarized along a proximal–distal axis (reviewed in Azimzadeh & Marshall, 2010; Gönczy & Hatzopoulos, 2019). In the proximal region lies a likewise symmetrical cartwheel usually ~100 nm in length, which is critical for scaffolding the onset of centriole assembly (reviewed in Guichard *et al*, 2018; Hirono, 2014). In transverse view, the cartwheel is characterized by a central hub from which emanates 9 spokes that extend toward peripherally located microtubule triplets.

The SAS-6 family of proteins is thought to constitute the principal building block of the cartwheel and is essential for its formation across systems (Dammermann *et al*, 2004; Leidel *et al*, 2005; Kilburn *et al*, 2007; Kleylein-Sohn *et al*, 2007; Nakazawa *et al*, 2007; Rodrigues-Martins *et al*, 2007; Strnad *et al*, 2007; Yabe *et al*, 2007; Culver *et al*, 2009; Jerka-Dziadosz *et al*, 2010). SAS-6 proteins contain an N-terminal globular head domain, followed by a ~45 nm long coiled-coil and a C-terminal region predicted to be unstructured (Dammermann *et al*, 2004; Leidel *et al*, 2005; van Breugel *et al*, 2011; Kitagawa *et al*, 2011). *In vitro*, SAS-6 proteins readily homodimerize through their coiled-coil moiety; such homodimers can undergo higher order oligomerization through an interaction between neighboring head domains (van Breugel *et al*, 2011; Kitagawa *et al*, 2011; Nievergelt *et al*, 2018). Ultimately, this results in the formation of a SAS-6 ring with a central hub harboring 18 juxtaposed head domains, from which emanate 9 paired coiled-coils that extend peripherally. Such ring oligomers are ~23 nm in diameter and bear striking resemblance with a transverse section of the cartwheel observed in cells. Moreover, recombinant *Chlamydomonas reinhardtii* SAS-6 (CrSAS-6) possesses the ability not only to self-assemble into ring oligomers, but also to undergo stacking of such entities, together generating a structure akin to the cartwheel present in the cellular context (Guichard *et al*, 2017).

Additional features of cartwheel architecture have been unveiled through cryo-electron tomography (cryo-ET) of centrioles purified

1 Swiss Institute for Experimental Cancer Research (ISREC), School of Life Sciences, Swiss Federal Institute of Technology Lausanne (EPFL), Lausanne, Switzerland
2 Interdisciplinary Centre for Electron Microscopy (CIME), Swiss Federal Institute of Technology Lausanne (EPFL), Lausanne, Switzerland
3 Department of Cell Biology, University of Geneva, Sciences III, Geneva, Switzerland
 *Corresponding author. Tel: +41 21 6930711; E-mail: pierre.gonczy@epfl.ch
 †These authors contributed equally to this work

from *Trichonympha* (Guichard *et al*, 2012, 2013). Three closely related species of these unicellular symbiotic flagellates, *T. campanula*, *T. collaris,* and *T. sphaerica,* referred to collectively as *T.* spp., populate the gut of *Zootermopsis* damp wood termites (Tai *et al*, 2013). *T.* spp. centrioles are particularly well suited for sub-tomogram averaging (STA) of cryo-ET specimens because they harbor an exceptionally long cartwheel-bearing region, reaching several microns (Gibbons & Grimstone, 1960; Guichard *et al*, 2012, 2013). STA of purified *T.* spp. centrioles yielded a ~34 Å map (Fourier Shell Correlation (FSC) criterion 0.143), which established that the cartwheel comprises stacks of ring-containing elements bearing a central hub from which emanate spokes. Suggestively, a 9-fold symmetrical SAS-6 ring generated computationally from the crystal structure of the CrSAS-6 head domain plus the first 6 heptad repeats of the coiled-coil (CrSAS-6[6HR]) could be fitted in the hub of this STA map (Guichard *et al*, 2012). However, some vertical hub densities remained unaccounted for upon such fitting, raising the possibility that additional components are present. In addition, the *T.* spp. STA map revealed 9-fold symmetrical cartwheel inner densities (CID) inside the hub proper, with contacts between hub and CID occurring where the fitted CrSAS-6[6HR] head domains interact with one another (Guichard *et al*, 2013).

The *T.* spp. STA map uncovered a vertical periodicity of ~8.5 nm between spoke densities emanating from the hub (Guichard *et al*, 2012, 2013). Two such emanating densities merge with one another as they extend toward the periphery, where the vertical spacing between merged entities is hence of ~17 nm. There, spokes abut a pinhead structure that bridges the central cartwheel with peripheral microtubule triplets. The STA map also revealed the architecture of the A-C linker, which connects the A-microtubule from a given triplet with the C-microtubule of the adjacent one. Interestingly, both pinhead and A-C linker are polarized along the proximal–distal centriole axis (Guichard *et al*, 2013, 2020). Given that the centrally located hub and CID were not noted at the time as being polarized, this led to the suggestion that the pinhead and the A-C linker might be critical for imparting polarity to the entire organelle (Guichard *et al*, 2013). Further cryo-ET analysis of procentrioles from *Chlamydomonas* and mammalian cells established that aspects of A-C linker architecture are evolutionarily conserved, including the attachment points on the A- and C-microtubules; moreover, novel features were revealed, such as a vertical crisscross pattern for the A-C linker in *Chlamydomonas* (Greenan *et al*, 2018; Li *et al*, 2019). Whether the central elements of the cartwheel, including the CID, are likewise conserved beyond *T.* spp. is unclear.

Considering that the earlier work in *T.* spp. was conducted without direct electron detector and that software improvements have occurred since, we sought to achieve a higher resolution map of the *T.* spp. cartwheel-bearing region. Moreover, to explore the evolutionarily conservation of cartwheel architecture, we investigated two other flagellates living in the gut of termites that might likewise harbor long cartwheels well suited for STA.

# Results

### Exceptionally long cartwheel region in *Trichonympha agilis*

We set out to obtain a high-resolution STA map of the native cartwheel in *T.* spp. Moreover, we likewise aimed at investigating *Trichonympha agilis*, a symbiotic flagellate that lives in the gut of the Japanese termite *Reticulitermes speratus* (Ohkuma & Kudo, 1998). This choice was guided by the fact that transcriptomic and genomic information is being assembled in *T. agilis* (Yuichi Hongoh, Tokyo Institute of Technology, Japan, personal communication), which will be instrumental to map proteins onto the STA map should sub-nanometer resolution be reached in the future.

As shown in Fig 1A, *T. agilis* bears a large number of flagella, which stem from similarly numerous centrioles inserted below the plasma membrane (Kubai, 1973). Many of these flagella are tightly packed in a region called the rostrum located at the cell anterior (Fig 1A, arrow). To determine the length of the cartwheel-bearing region of *T. agilis* centrioles, cells were resin-embedded and analyzed by transmission electron microscopy (TEM), focusing on the rostral region. Longitudinal views established that the cartwheel-bearing region is ~2.3 μm in length on average (Fig 1B, pink line; SD = 0.36 μm, $N = 8$). This is less than the ~4 μm observed in *T.* spp. (Guichard *et al*, 2013), yet over 20 times the size of the ~100 nm cartwheel in centrioles of most systems, including *Chlamydomonas reinhardtii* and *Homo sapiens* (Guichard *et al*, 2010, 2017; O'Toole & Dutcher, 2014). In addition, we found that the centriole distal region devoid of cartwheel is ~0.4 μm in *T. agilis* (Fig 1B, white line; SD = 0.07 μm, $N = 6$), similar to its dimensions in other systems (Guichard *et al*, 2013; Le Guennec *et al*, 2020).

We also analyzed transverse sections of resin-embedded *T. agilis* centrioles using TEM. As shown in Fig 1C, we found the characteristic features of the cartwheel-bearing region, including a central hub from which emanate 9 spokes that extend toward peripheral microtubule triplets. In addition, we noted the presence of the pinhead and the A-C linker, as well as of the triplet base connecting these two elements (Gibbons & Grimstone, 1960; Vorobjev & Chentsov, 1980), which is more apparent in the circularized and symmetrized image (Fig 1C).

Overall, given the presence of a long cartwheel-bearing region, we conclude that *T. agilis* also provides a suitable system to investigate the architecture of the proximal part of the centriole using cryo-ET and STA.

### Novel features revealed by improved STA of *Trichonympha* centrioles

Using a direct electron detector, we acquired tilt series of purified *T.* spp. centrioles, focusing on the proximal cartwheel-bearing region (Appendix Fig S1A and B), followed by tomogram reconstruction and STA (Fig 2A–E; for all datasets, see Appendix Fig S1C–F for raw tomograms, as well as Appendix Fig S1G–J and Appendix Table S1 for processing pipeline). For the central cartwheel, we achieved a local resolution ranging from ~16 Å to ~40 Å (Appendix Fig S2A), with a global resolution of ~24 Å (FSC criterion 0.143; Appendix Fig S2B; see Appendix Figs S3–S5 for resolution of all other STA maps, which have been deposited in EMDB).

Using line scans on 2D projections of the STA map, we determined the *T.* spp. hub diameter to be ~23 nm (Fig 2A and B), in line with previous work (Guichard *et al*, 2012, 2013). Importantly, the improved resolution achieved here enabled us to uncover novel features in the central cartwheel of the *T.* spp. centriole. Of particular interest, we discovered that the position of the CID is polarized along the proximal–distal centriole axis with respect to the hub and

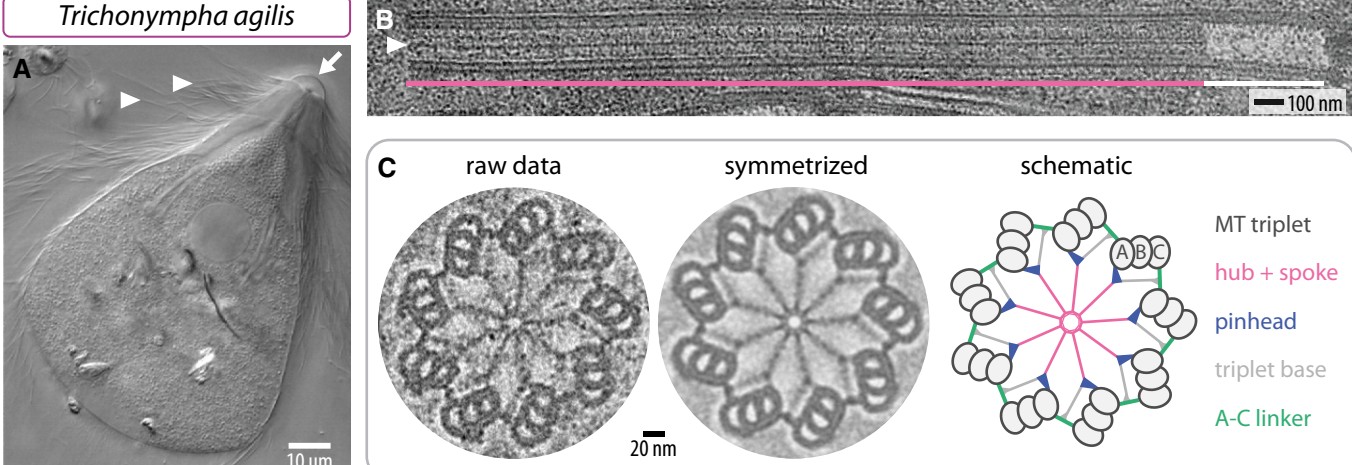

**Figure 1. Exceptionally long centriolar cartwheel in *T. agilis*.**

A  Differential interference contrast micrograph of live *T. agilis* cell. The arrow points to the cell anterior, where the rostrum is located; arrowheads point to some of the flagella.

B  Transmission electron micrographs of *T. agilis* centriole embedded in resin—longitudinal view; the hub (arrowhead) is visible in the cartwheel-bearing region (pink line), but not in the distal region (white line).

C  Transmission electron micrographs of *T. agilis* centriole embedded in resin in transverse view from distal end (left) and corresponding image circularized and symmetrized with the CentrioleJ plugin (middle), with schematic indicating principal architectural elements (right).

the spokes that emanate from it (Fig 2C and D). This is apparent from vertical intensity profiles of longitudinal views, which show that the CID is positioned distal to the center of the hub density, which itself appears to be elongated in the vertical direction (Fig 2C). Occasionally, two units can be discerned within one hub density (Fig 2C, double peaks in blue intensity profile and corresponding dashed lines), a point that will be considered further below. Such double units were not recognized previously, presumably owing to the lower resolution map (Guichard *et al*, 2012, 2013). Moreover, we found densities that vertically bridge successive hub elements (Fig 2D, arrows). The polarized location of the CID unveiled here is also apparent with respect to where spoke densities emerge from the hub (Fig 2E, arrow). In addition, we identified discontinuous densities in the center of the CID (Fig 2B and D).

We likewise analyzed the central cartwheel in *T. agilis*, observing variations along the centriole axis in 2D longitudinal views (Appendix Fig S1D). Focused 3D classification of sub-volumes indeed uncovered two classes (Fig 2F–J; Fig EV1), corresponding to 55 and 45% of sub-volumes, which can occur within the same centriole (Fig EV1F). We found that the central cartwheel STA map of both *T. agilis* classes exhibits many similarities with that of *T.* spp. Thus, the CID is present and spoke densities emanate from a hub ~23 nm in diameter (Figs 2F and G, and EV1A and B). Moreover, vertical densities bridging successive hub elements are also present in both *T. agilis* classes (Figs 2I and EV1D, arrows). Furthermore, we found that the CID is also polarized along the proximal–distal centriole axis, being distal with respect to the hub in both *T. agilis* classes, as evidenced from vertical intensity profiles (Figs 2H and I and EV1C and D), as well as from the location of the CID relative to where spoke densities emerge from the hub (Fig 2J, arrow; Fig EV1D, arrowhead). In the *T. agilis* 55% class, like in *T.* spp., hub densities are elongated in the vertical direction and can

be sometimes discerned as two units (Fig 2H, double peaks in light pink intensity profile and corresponding dashed lines). The presence of such double hub units next to the CID is more apparent in the 45% class, where they also exhibit a slight offset relative to the vertical axis (Fig EV1C, white dashed line), a point considered further below. In addition, we found in this class that double hub units alternate with single hub units that do not have a CID in their vicinity (Fig EV1C, dashed pink arrow, Fig EV1D), an absence noticeable also in raw tomograms (Appendix Fig S1D, empty arrowheads) and verified in 3D top views (Fig EV1A and B, full circle). Moreover, we found that spoke densities emanating from single hub units are thinner than those stemming from double hub units (Fig EV1D). The plausible origin of alternating double and single hub units in the 45% *T. agilis* class will be considered below. We noted also that the 45% sub-volumes exhibit slight variations in the spacing between double and single hub units (Fig EV1E, 25 and 20% sub-classes).

Taken together, our findings establish that *T.* spp. and *T. agilis* central cartwheel architecture shares many features, including a polarized CID position.

## Comparative STA of peripheral centriole elements in *Trichonympha*

We also investigated peripheral elements in the proximal region of *T.* spp. and *T. agilis* centrioles. To this end, we extracted peripheral sub-volumes from the tomograms and generated for each species three maps using STA centered either on the microtubule triplet, the pinhead, or the A-C linker (Fig 3A and E).

For the microtubule triplet, the resulting analysis revealed a characteristic centriolar architecture. Thus, the A-microtubule bears 13 protofilaments, the B-microtubule 10 protofilaments proper, with 3

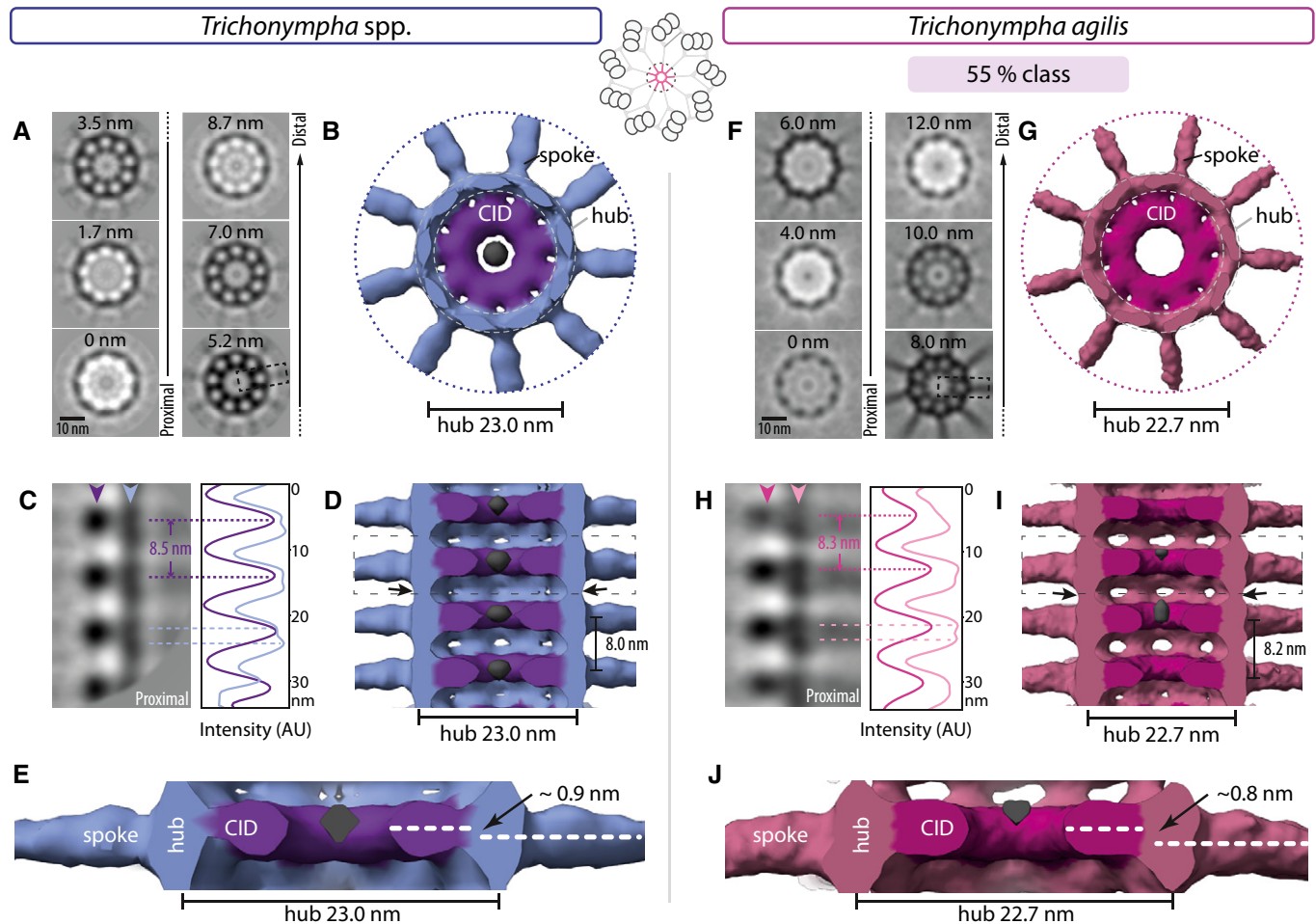

**Figure 2. Conserved architecture and polarity in *T.* spp. and *T. agilis* cartwheel.**

A   Transverse 2D slices through *T.* spp. central cartwheel STA at indicated height from proximal (0 nm) to distal (8.7 nm). The dashed box indicates corresponding region shown in (C). Schematic on top center illustrates the area used to generate the 3D maps of the central cartwheel.

B   Transverse view of *T.* spp. central cartwheel STA 3D map; 9 spoke densities emanate from the hub, and the CID is present within the hub. The diameter of the hub is 23.0 ± 0 nm (*N* = 3; here and thereafter in the figure legends, *N* corresponds to 2D measurements from STA). An electron-dense structure is present inside the CID (gray), which is also visible to some extent before symmetrizing. Note that the coloring of elements in this and other figure panels is merely illustrative and not meant to reflect underlying molecular boundaries.

C   2D longitudinal view of *T.* spp. central cartwheel STA delineated by a dashed box in (A). Arrowheads denote position of line scans along the vertical axis at the level of the CID (purple) and hub (blue), with corresponding pixel intensities in arbitrary units (AU). Plot profiles are shifted horizontally relative to each other for better visibility. Some maxima are highlighted with dashed lines; the average distance between two CID elements is 8.5 ± 0.2 nm (*N* = 4). Note that hub densities are elongated in the vertical direction, resulting in broad peak profiles where two maxima can be discerned (dashed blue lines).

D   Longitudinal view of *T.* spp. central cartwheel STA. The average distance between emanating spokes is 8.0 ± 1.5 nm (*N* = 2); these are measurements on STA and are within the error of the CID periodicities reported in (C); the same applies for other measurements hereafter. Discontinuous densities in the center of the CID (gray) are visible, as well as densities that vertically bridge successive hub elements (arrows). The dashed box is shown magnified in (E). Proximal is down in this and all other figure panels showing STA longitudinal views.

E   The CID axis is located distal relative to the axis of spoke densities, corresponding to an average shift of 0.9 ± 0.7 nm (*N* = 3).

F   Transverse 2D slice through *T. agilis* central cartwheel STA at indicated height from proximal (0 nm) to distal (12 nm). The dashed box indicates corresponding region shown in (H).

G   Transverse view of *T. agilis* central cartwheel STA 3D map; 9 spoke densities emanate from the hub, and the CID is present within the hub, which has a diameter of 22.7 ± 0.2 nm (*N* = 3).

H   2D longitudinal view of *T. agilis* central cartwheel STA delineated by a dashed box in (F). Arrowheads denote position of line scans along the vertical axis at the level of the CID (dark pink) and hub (light pink), with corresponding pixel intensities in arbitrary units (AU). Plot profiles are shifted horizontally relative to each other for better visibility. Some maxima are highlighted with dashed lines; the average distance between two CID elements is 8.3 ± 0.5 nm (*N* = 9). Note that hub densities are elongated in the vertical direction, resulting in broad peak profiles where two maxima can be discerned (dashed light pink lines).

I    Longitudinal view of *T. agilis* central cartwheel STA. The average distance between emanating spokes is 8.2 ± 0.5 nm (*N* = 9). Discontinuous densities in the center of the CID (gray) are visible at this lower contour level compared to (G). Note densities vertically bridging successive hub elements (arrows). Boxed area is shown magnified in (J).

J    The CID axis is located distal relative to the axis of spoke densities, corresponding to an average shift of 0.8 ± 0.3 nm (*N* = 9).

extra ones shared with the A-microtubule, whereas the C-microtubule exhibits a similar organization as the B-microtubule (Fig 3B and F). Moreover, we detected prominent densities corresponding to microtubule inner proteins (MIPs). In both species, we found a MIP located along protofilament A9, close to protofilament A10 (Fig 3B and F, empty arrowhead). A MIP was discovered at this location in ciliary axonemes (Nicastro *et al*, 2006) and was also observed in centrioles of *T.* spp., *Chlamydomonas* and mammalian cells (Guichard *et al*, 2013; Greenan *et al*, 2018, 2020; Li *et al*, 2019). In addition, we observed a MIP along protofilament A5 in *T.* spp. (Fig 3B, chevron), which is visible only at low threshold in *T. agilis* and positioned like MIP1 in *Chlamydomonas* centrioles as well as axonemes (Nicastro *et al*, 2006; Li *et al*, 2019). In both *Trichonympha* species, we also detected additional densities or microtubule-associated proteins (MAPs) on the microtubule exterior between the A-B and B-C inner junctions (Fig 3B and F, arrowheads), as described also in *Chlamydomonas* and mammalian centrioles (Greenan *et al*, 2018; Li *et al*, 2019).

For both *T.* spp. and *T. agilis*, we next conducted STA centered on the pinhead or the A-C linker to uncover features in these peripheral elements. We thus found that the pinhead connects to the A3 protofilament in both species (Fig 3C and G), in line with previous observations in *T.* spp. and in other systems (Guichard *et al*, 2013; Greenan *et al*, 2018; Li *et al*, 2019). Moreover, we found that the pinhead is polarized in a similar manner in *T.* spp. and *T. agilis* (Fig 3C and G), with the two pinfeet moieties PinF1 and PinF2 pointing proximally from the A3 protofilament, as reported previously for *T.* spp. (Guichard *et al*, 2013, 2020). We found also that the spacing between PinF1 and PinF2 elements is ~8.6 nm and ~7.9 nm in *T.* spp., whereas it is ~8.4 nm and ~8.4 nm in *T. agilis* (Fig 3C and G), compatible with power spectra of 2D class averages considering the standard deviation of the measurements (Appendix Fig S2I, Appendix Fig S3L).

Similarities between the two species are also apparent in the A-C linker that bridges neighboring MT triplets. Thus, we found that the *T.* spp. A-C linker connects protofilament A8/A9 from one triplet with protofilaments C9/C10 of the adjacent triplet (Fig 3D), furthering the mapping of these connections compared to previous work (Guichard *et al*, 2013). As shown in Fig 3H, we found that the A-C linker in *T. agilis* has a similar architecture. To generate an overview of the entire proximal region of the *T. agilis* centriole, comprising hub, spoke, pinhead, and A-microtubule, we conducted STA centered on the spokes for both 55 and 45% classes (Fig EV2A–H). Such maps have a lower resolution due to the larger box size, the binning and the absence of radial symmetrization, but nevertheless uncover a concerted proximal–distal polarity of several elements. First, the CID is positioned distally within double hub units. Second, spoke densities exhibit a slight asymmetry in their tilt angle, with the proximal spoke being more tilted. Moreover, these two polarized features present centrally are connected with the polarized pinhead and A-C linker present peripherally (Fig EV2C, D, G and H).

Overall, we conclude that peripheral components of the cartwheel are also generally conserved between the two *Trichonympha* species and exhibit concerted polarity with central elements along the proximal–distal centriole axis.

## Diversity of cartwheel architecture in *Teranympha* cartwheel

The gut of *R. speratus* termites contains another symbiotic flagellate, namely *Teranympha mirabilis* (Koidzumi, 1921; Noda *et al*, 2018), which we found also to harbor numerous centrioles and associated flagella (Fig EV3A). TEM analysis of resin-embedded specimens established that the cartwheel-bearing region in such centrioles is ~1.1 μm on average (Fig EV3B, green line; $N = 9$, SD = 0.06 μm), whereas the distal region averages ~0.5 μm (Fig EV3B, black line; $N = 9$, SD = 0.08 μm). Transverse sections of

**Figure 3. Architecture of peripheral elements in *T.* spp. *and T. agilis*.**

A (Top) 2D slice through STA transverse view of *T.* spp. microtubule triplet, with insets showing position of pinhead (dashed green box) and A-C linker (dashed red box). (Bottom) Longitudinal 2D slice of STA centered on the pinhead (left) or A-C linker (right). Schematic on top center illustrates the different areas used to generate maps of the microtubule triplets (B and F), pinhead (C and G), and A-C linker (D and H).

B Transverse view of *T.* spp. microtubule triplet STA. Microtubule protofilament numbers are indicated, as are the pinhead and A-C linker (only the C-link is visible; the A-link lies on the edge of the volume and is thus less well resolved in this STA—for better view, see STA centered on A-C linker in (D)). Prominent microtubule inner densities within the A-microtubule are highlighted (empty arrowhead next to A9, chevron next to A5), as are additional external densities at the A-B and B-C inner junctions (black arrowheads). Double arrowheads point to viewing direction in indicated panels.

C Longitudinal view of *T.* spp. STA centered on the pinhead from the viewing point indicated in (B). The pinfeet (PinF1 and PinF2) and pinbody (PinB) are indicated, as are microtubule protofilaments A3 and A4. The average distance between pinfeet elements is 8.6 ± 0.4 nm and 7.9 ± 0.4 nm ($N = 3$ each). Corresponding transverse views are shown below, illustrating the connection of PinF2 with protofilament A3.

D Longitudinal view of *T.* spp. STA centered on the A-C linker from the viewing point indicated in (B). Microtubule protofilaments A8/9 and C9/C10 of two adjacent triplets are indicated, as are the connected A- and C-links. The average distance between A- and C-links is 8.4 ± 0.4 nm and 8.4 ± 0.3 nm (both $N = 6$). Corresponding transverse views are shown below; chevrons point to connection.

E (Top) 2D slice through STA transverse view of *T. agilis* microtubule triplet, with insets showing position of pinhead (dashed green box) and A-C linker (dashed red box). (Bottom) Longitudinal 2D slice of STA centered on the pinhead (left) or A-C linker (right).

F Transverse view of *T. agilis* microtubule triplet STA. Microtubule protofilament numbers are indicated, as are the pinhead and A-C linker (only the C-link is visible; the A-link lies on the edge of the volume and is thus less well resolved in this STA—for better view, see STA centered on A-C linker in (H)). Prominent microtubule inner densities (MIPs) within the A-microtubule are highlighted (empty arrowhead next to A9), as are additional external densities at the A-B and B-C inner junctions (black arrowheads). Double arrowheads point to viewing direction in indicated panels.

G Longitudinal view of *T. agilis* STA centered on the pinhead from the viewing point indicated in (F). The pinfeet (PinF1 and PinF2) and pinbody (PinB) are indicated, as are microtubule protofilaments A3/A4. The average distance between pinfeet elements is 8.4 ± 0 nm in each case (both $N = 3$). Corresponding transverse views are shown below, illustrating the connection of PinF2 with protofilament A3.

H Longitudinal view of *T. agilis* STA centered on the A-C linker from the viewing point indicated in (F). Microtubule protofilaments A8/9 and C9/C10 of two adjacent triplets are indicated, as are the connected A- and C-links. The average distance between A- and C-links is 8.4 ± 0.3 nm and 8.4 ± 0.9 nm ($N = 6$ and $N = 5$, respectively). Corresponding transverse views are shown below; chevrons point to connection; the connection of the A-link with A9 is only partially visible in the transverse view at this height, as indicated by the dashed chevron.

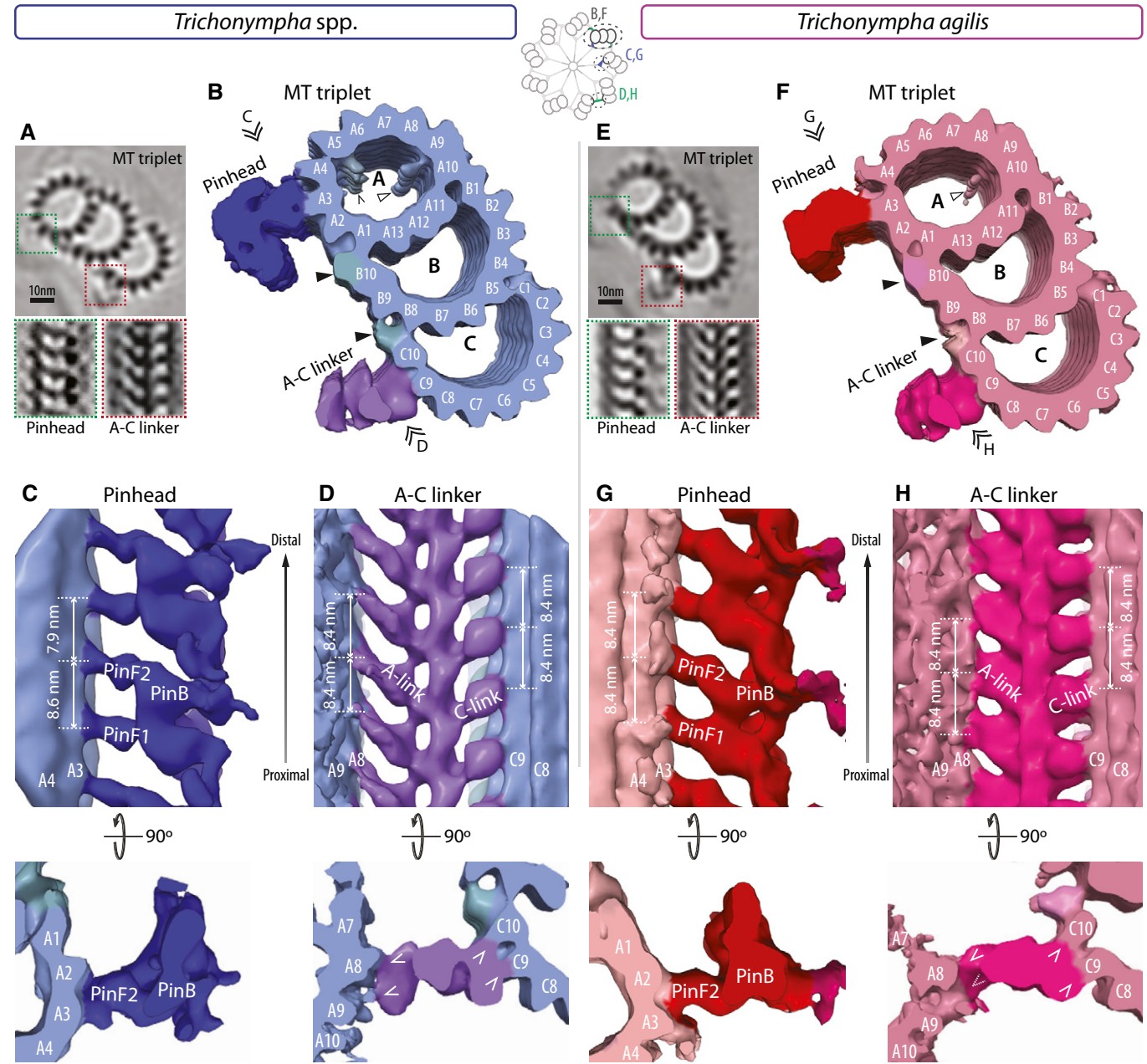

**Figure 3.**

the cartwheel-bearing region revealed the characteristic hub and spokes connected through a pinhead to peripheral microtubule triplets, which are joined by an A-C linker, whereas the triplet base is poorly visible (Fig EV3C).

We conducted cryo-ET of purified *T. mirabilis* centrioles, acquiring tilt series of the entire cartwheel-bearing region followed by tomogram reconstruction and STA. We again generated separate maps for the central cartwheel, the microtubule triplet, the pinhead, and the A-C linker. Focused 3D classification of the central cartwheel region yielded two classes representing 64 and 36% of subvolumes, which can occur within the same centriole (Figs 4A–G and EV3E). Similarly to the two *Trichonympha* species, a hub ~23 nm in diameter is present in both *T. mirabilis* classes, with spoke densities

emanating from it (Fig 4A–C). Interestingly, we found that the architecture of the central cartwheel in *T. mirabilis* differs slightly from that in *Trichonympha*. Indeed, in both *T. mirabilis* classes, we uncovered a filamentous structure ~7 nm in diameter present along the entire cartwheel-bearing region inside the hub, which we dubbed filamentous cartwheel inner density (fCID) (Fig 4A–G). The fCID is also apparent in transverse views of resin-embedded centrioles and symmetrized tomogram slices (Fig EV3C and D). Moreover, the fCID is consistently detected in raw tomograms (Appendix Fig S1E), as well as in non-symmetrized STA comprising larger volumes centered on the spokes (Fig EV2I–P).

As shown in Figure 4D–G, the two central cartwheel *T. mirabilis* classes differ in hub architecture, as revealed by vertical line profile

intensity measurements. In the 64% class, a periodicity of ~8.4 nm is apparent between hub densities, each consisting seemingly of a single vertically elongated unit (Fig 4D and F). By contrast, in the 36% class, each hub density comprises a double unit (Fig 4E and G). Such double hub units exhibit a peak-to-peak distance of ~3.2 nm and are separated from the adjacent double hub unit by ~5.2 nm (Fig 4E). The sum of the two distances, namely 8.4 nm is equivalent to that observed in the 64% class. Moreover, the periodicity at the level of emerging spoke densities is similar in the two classes (Fig 4F and G). Furthermore, as in the 45% *T. agilis* class, we observed that every other double hub unit in the 36% *T. mirabilis* class exhibits a slight offset relative to the vertical axis (Fig 4E, white dashed lines).

Analysis of the peripheral STA microtubule triplet map of the *T. mirabilis* centriole uncovered the canonical protofilament configuration for A-, B-, and C-microtubules (Fig 4H and I). As in *Trichonympha*, we detected additional densities on the external side of the microtubules between A-B and B-C inner junctions (Fig 4I, arrowheads). The previously described C-stretch that extends from protofilament C1 in *T.* spp. (Guichard *et al*, 2013) is also observed in *T. mirabilis* (Fig 4I, chevron), while prominent MIPs are not detected at the selected threshold.

The map generated by centering on the pinhead revealed a connection with the A3 protofilament, with PinF1 and PinF2 being separated by 8.4 nm (Fig 4J), as suggested also by power spectra of 2D class averages (Appendix Fig S4L). Moreover, we found that the *T. mirabilis* A-C linker architecture is similar to that in the two *Trichonympha* species, with anchoring to microtubules at protofilaments A9 and C9/C10 (Fig 4K).

Overall, these findings indicate that *T. mirabilis* peripheral elements share many conserved features with those in *Trichonympha*, as does the central cartwheel, apart from the discontinuous CID being replaced by the seemingly continuous fCID in *T. mirabilis*.

## Comparative analysis of SAS-6 ring stacking mode

We set out to investigate how SAS-6 rings may fit into the central cartwheel STA maps of the three species. Genomic or transcriptomic information is not available for *T.* spp. and *T. mirabilis* at present. Thus, in the absence of molecular information for SAS-6 in these two species, as well as of structural information for *T. agilis* SAS-6, and given that SAS-6 crystal structures from a wide range of organisms exhibit extensive similarities, we employed CrSAS-6 as a model instead. We used homodimers of CrSAS-6[6HR], the longest CrSAS-6 fragment with a determined crystal structure, which can self-assemble into 9-fold symmetrical rings ~23 nm in diameter and 4.2 nm in height (Kitagawa *et al*, 2011).

Given the presence of two units per hub density in the *T. mirabilis* 36% class, and also probably in both *Trichonympha* species, we computationally assembled CrSAS-6[6HR] into single rings, as well as into directly superimposed double rings in register. We then used rigid-body fitting to score the goodness of fit of both types of ring assemblies with the STA maps (Appendix Table S2). Starting with the *T. mirabilis* 36% class, we found that whereas a single CrSAS-6[6HR] ring fits in this map, additional hub densities remain unaccounted for in this case (Fig 5A, arrowheads). By contrast, the double ring configuration fills the hub density to

a larger extent (Fig 5B), yielding a better overlap score (Appendix Table S2). However, coiled-coil moieties extend slightly outside the spoke densities in this case, potentially because of the coiled-coil bending *in vivo* or due to species-specific features. Moreover, rigid-body fitting of CrSAS-6[6HR] homodimers confirmed that two directly superimposed dimers provide a better fit for the *T. mirabilis* 36% class, a fit that also revealed a slight offset between them (Fig EV4A). Next, we fitted computationally assembled single and double CrSAS-6[6HR] rings in the *T. mirabilis* 64% class map, finding again that head domains of double rings can be readily accommodated in the hub density, although one coiled-coil moiety clearly extends outside spoke densities in this case (Fig EV4B). Taken together, these observations are compatible with the possibility that SAS-6 rings can directly stack on top of one another *in vivo*.

To explore whether direct SAS-6 ring stacking might occur also in *Trichonympha*, we likewise performed fitting of computationally assembled CrSAS-6[6HR] single and double rings. We found again that in all cases directly superimposed double rings can readily be accommodated in the hub densities with improved goodness of fit compared to single rings (Figs 5C and D, and EV4C and D; Appendix Table S2). The *T. agilis* 45% class, which comprises alternating double and single hub elements, presents a particularly interesting case. As anticipated, double rings could be placed in the elongated hub density comprising two units, yet a single ring could be accommodated in the thinner individual hub unit (Fig 5C and D; Appendix Table S2).

Prompted by the slight offset observed between two SAS-6 rings upon fitting CrSAS-6[6HR] homodimers in the *T. mirabilis* 36% class (see Fig EV4A), as well as the structural complementarity between superimposed SAS-6 rings following such an offset, we computationally assembled a double ring with a ~6.5° rotational offset imposed between ring pairs, which allowed the two rings to come closer to one another vertically by 0.4 nm (Fig EV5B, compare with Fig EV5A; Materials and Methods). Manual fitting in the *T. mirabilis* 36% class map uncovered that such an offset double ring can be readily accommodated in the hub (Fig EV5C). A similar conclusion was drawn from fitting the offset double ring into the double hub density of the *T. agilis* 45% class (Fig EV5D). Such a slight offset between superimposed SAS-6 rings might explain the hub offset observed in the longitudinal view of the central cartwheel STA in both *T. mirabilis* 36% class and *T. agilis* 45% class (see Figs 4E and EV1C, white dashed lines). Furthermore, we reasoned that such an offset double ring might likewise affect the connected spoke densities. If spoke densities correspond in reality to two individual spokes slightly offset from one another, as expected from an offset double ring configuration, but which cannot be identified as individual units due to the resolution limit, then the shape of the spoke densities should be elliptical when viewed end on. To investigate this possibility, we unwrapped the central cartwheel maps to obtain a complete view at the level of the spokes (Fig EV5E). Importantly, this uncovered the expected elliptical shape, revealing spoke offset in the *T. mirabilis* 36% class (Fig EV5F, dashed line), as well as in both *T. agilis* classes (Fig EV5G) and in *T.* spp (Fig EV5H). By contrast, no spoke offset was apparent in the *T. mirabilis* 64% class (Fig EV5F).

Overall, these findings lead us to propose that pairs of SAS-6 rings directly stack in the cellular context and can do so with an offset (see Discussion).

We investigated further the possibility that SAS-6 rings directly stack onto one another using a cell-free assay with a recombinant CrSAS-6 protein containing the globular head domain and the entire coiled-coil (referred to as CrSAS-6[NL]). It has been shown previously that CrSAS-6[NL] bearing 6xHis- and S-tag moieties can self-organize into stacks thought to exhibit ~8.5 nm periodicity between spokes, based on the analysis strictly of top views (Guichard *et al*, 2017). We investigated this question anew, now analyzing stacking periodicity of untagged purified CrSAS-6[NL] protein. Moreover, we conducted STA of side views of stacked assemblies, thus allowing us to measure the underlying periodicities with a global resolution of ~25 Å (Fig 5E–I, Appendix Fig S5). Importantly, we found that such *in vitro* assembled CrSAS-6[NL] stacks exhibit a periodicity of ~4.5 nm that can each accommodate one computationally assembled single ring of CrSAS-6[6HR] (Fig 5G–I). Moreover, we found that the spokes of successive *in vitro* assembled rings are almost in register (Fig EV5I).

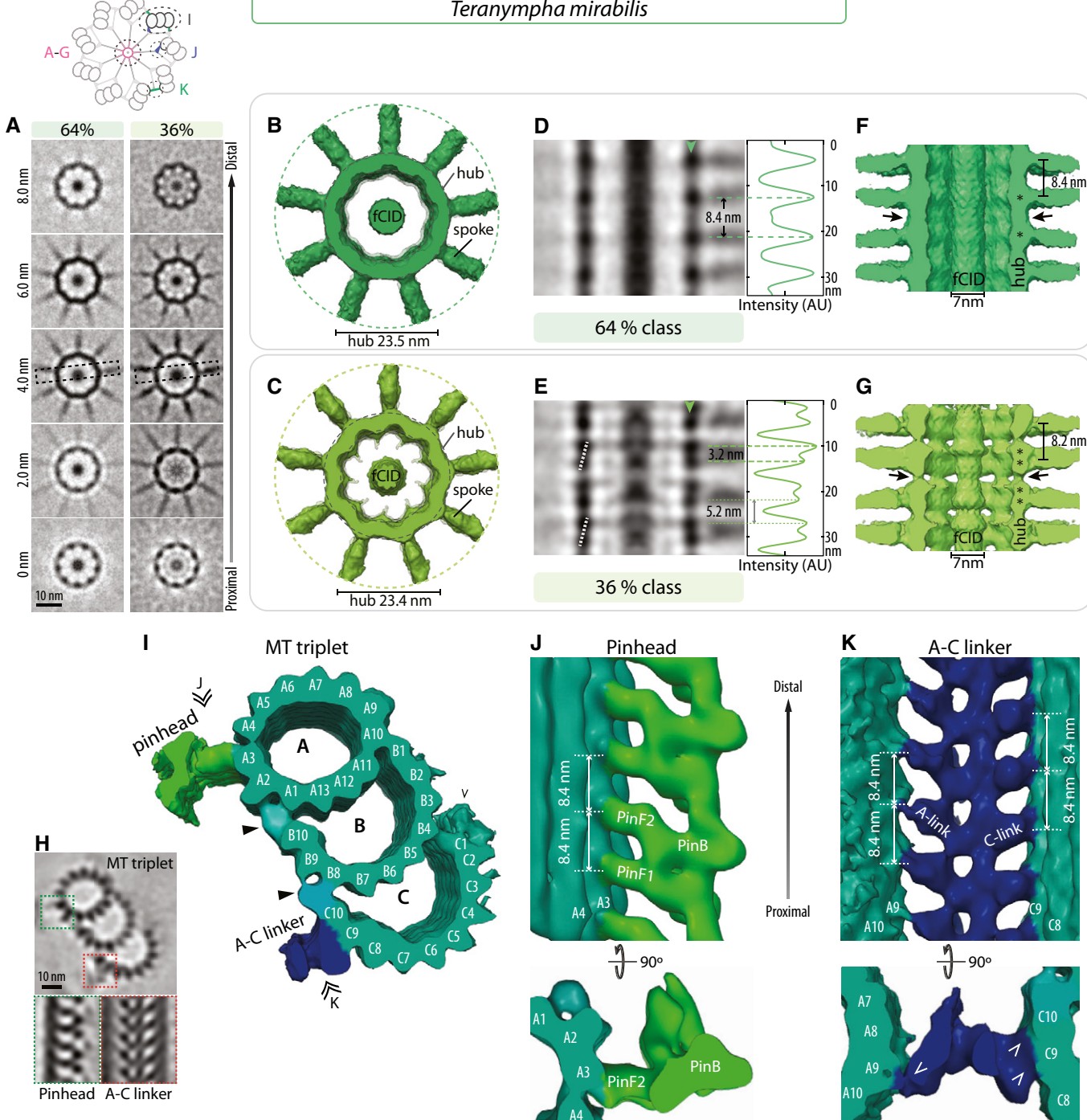

**Figure 4.**

**Figure 4.  Novel architectural features in *T. mirabilis* cartwheel.**

A   Transverse 2D slices through STA comprising 64 and 36% of sub-volumes of *T. mirabilis* central cartwheel at indicated height from proximal (0 nm) to distal (8.0 nm). The dashed boxes in the 4.0 nm slices indicate regions shown in (D and E). Note filamentous cartwheel inner densities (fCID) in the center of the hub. Schematic on top illustrates the different areas used to generate 3D maps of the central cartwheel (A-G), MT triplets (I), pinhead (J), and A-C linker (K).

B, C   Transverse view of *T. mirabilis* central cartwheel STA of 64% (B) and 36% (C) classes, with corresponding hub diameters of 23.5 ± 0.2 nm and 23.4 ± 0.5 nm (both *N* = 3). In both classes, the fCID is visible and 9 spokes emanate from the hub.

D, E   2D longitudinal view of *T. mirabilis* central region of the cartwheel STA 64% (D) and 36% (E) classes in the region delineated by dashed boxes in (A). Arrowheads denote position of line scans along vertical axis at the hub level, with corresponding normalized pixel intensities (in arbitrary units). Some maxima are highlighted by dashed lines. The average distance between hub units in the 64% class is 8.4 ± 0.6 nm (*N* = 8; D); in the 36% class, the distance between hub densities alternates between 3.2 ± 0.3 nm and 5.2 ± 0.3 nm and (both *N* = 9; E). Dashed white line in (E) indicates the offset between two superimposed hub units at the level of spoke densities, which occurs every other hub unit pair.

F, G   Longitudinal view of *T. mirabilis* cartwheel STA 64% (F) and 36% (G) classes. The average distance between spokes is 8.4 ± 0.6 nm (*N* = 8) in the 64% class and 8.2 ± 1.5 nm (*N* = 7) in the 36% class. Note in both cases the continuous fCID with ~7 nm in diameter inside the hub. Note also densities bridging successive hubs vertically (arrows). Asterisks denote positions of individual units apparent within hub densities. Note that the fCID was not always positioned in the geometrical center of the hub, suggestive of inherent flexibility.

H   (Top) 2D slice through STA transverse view of microtubule triplet in *T. mirabilis*, with insets showing pinhead (dashed green box) and A-C linker (dashed red box). (Bottom) Longitudinal 2D slice of STA centered on the pinhead (left) or A-C linker (right).

I   Transverse view of microtubule triplet STA in *T. mirabilis*. Microtubule protofilaments are indicated, as are the positions of the pinhead and the C-link (the A-link lies on the edge of the volume and is thus less well resolved in this STA—for better views, see STA centered on A-C linker in panel K). At this contour level, MIPs are not visible. Arrowheads indicate external densities at the A-B and B-C inner junctions; chevron indicates C-stretch. Double arrowheads point to viewing directions shown in (J, K).

J   Longitudinal view of *T. mirabilis* STA centered on the pinhead, from the viewing point indicated in (I). Location of pinhead consisting of pinfeet (PinF1 and PinF2) and pinbody (PinB) are indicated, as are microtubule protofilaments A3/A4. The average distance between pinfeet elements is 8.4 ± 0.5 nm in each case (both *N* = 5). Corresponding transverse view is shown below.

K   Longitudinal view of *T. mirabilis* STA centered on the A-C linker, from the viewing point indicated in (I). Microtubule protofilaments A8/A9 and C9/C10 of two adjacent triplets are indicated, as are the connected A- and C-links. The average distance between A- and C-links is 8.4 ± 0.9 nm (*N* = 6) and 8.4 ± 0.3 nm (*N* = 5), respectively. Corresponding transverse view is shown below; chevrons point to connections.

Taken together, these observations lead us to propose that direct stacking of SAS-6 rings may be an evolutionarily conserved feature of cartwheel assembly at the onset of centriole biogenesis.

# Discussion

The centriole is fundamental for numerous aspects of cell physiology and shows remarkable conservation across the major eukaryotic groups of life. Understanding the mechanisms that govern centriole assembly requires not only the identification of essential building blocks and their regulators, but also a detailed knowledge of organelle architecture. Our findings uncover conserved features in the cartwheel-bearing portion of the centriole. We find that the CID located within the hub is polarized along the proximal–distal centriole axis in both *Trichonympha* species and that a potentially related fCID is present in *T. mirabilis*. Moreover, we establish that hub densities can be fitted by directly superimposing 9-fold symmetrical SAS-6 double rings that exhibit a slight rotational offset between the two rings. Furthermore, we find that such an offset double ring is stacked in register with the next offset double ring. Overall, our work provides novel insights into the architecture of the centriolar cartwheel at the root of centriole biogenesis.

## Conserved features in cartwheel architecture

Centriole ultrastructure began to be explored in the 1950s when TEM of resin-embedded fixed specimens revealed its signature 9-fold radial symmetry and the chiral arrangement of microtubule triplets (Fawcett & Porter, 1954). More recent cryo-ET analysis uncovered the native architecture of the cartwheel-bearing portion of the centriole (Guichard *et al*, 2013; Li *et al*, 2019). Here, we set out to improve the resolution previously obtained with *T.* spp. and

to investigate whether features discovered in this species are present in other flagellates harboring unusually long cartwheels.

In the absence of fossil record or complete sequence information, it is not possible to assess sequence divergence between centriolar proteins in the species analyzed here, nor to date with precision the evolutionary times separating them. However, an approximation is provided in the case of *T.* spp. and *T. agilis* by the phylogenetic divergence between their respective hosts *Zootermopsis* spp. and *Reticulitermes speratus*, which are estimated to have shared their last common ancestor ~140 million years ago (Bucek *et al*, 2019). Given that *Trichonympha* are obligate symbiotic organisms, and considering that the two host species inhabit different continents (Thorne *et al*, 1993; Park *et al*, 2006), it is reasonable to postulate that *T.* spp. and *T. agilis* diverged at least this long ago. *Teranympha mirabilis* belongs to a different genus, and based on small subunit rRNA sequences it is much more distant from the *Trichonympha* species than they are from one another; however, the split between the two genera is thus far undated (Carpenter & Keeling, 2007; Noda *et al*, 2009, 2018; Ohkuma *et al*, 2009). Regardless of the exact evolutionary times separating these flagellates, our findings, together with those of a companion manuscript (Klena *et al*, 2020), establish that the CID is conserved between distant eukaryotic groups, suggesting shared evolutionary history and/or function (Fig 6A). The CID exhibits a 9-fold radial symmetry and connects with the hub approximately where neighboring SAS-6 homodimers interact with one another (Fig 6B), a suggestive location raising the possibility that the CID imparts or maintains the 9-fold symmetrical SAS-6 ring structure (Guichard *et al*, 2013). We discovered here that the CID is polarized along the proximal–distal centriole axis, and may thus also play a role in imparting or maintaining organelle polarity (Fig 6C). Furthermore, the CID exhibits an ~8.4 nm periodicity along the proximal–distal centriole axis, with no apparent continuity between two adjacent CID elements. This is in contrast to the

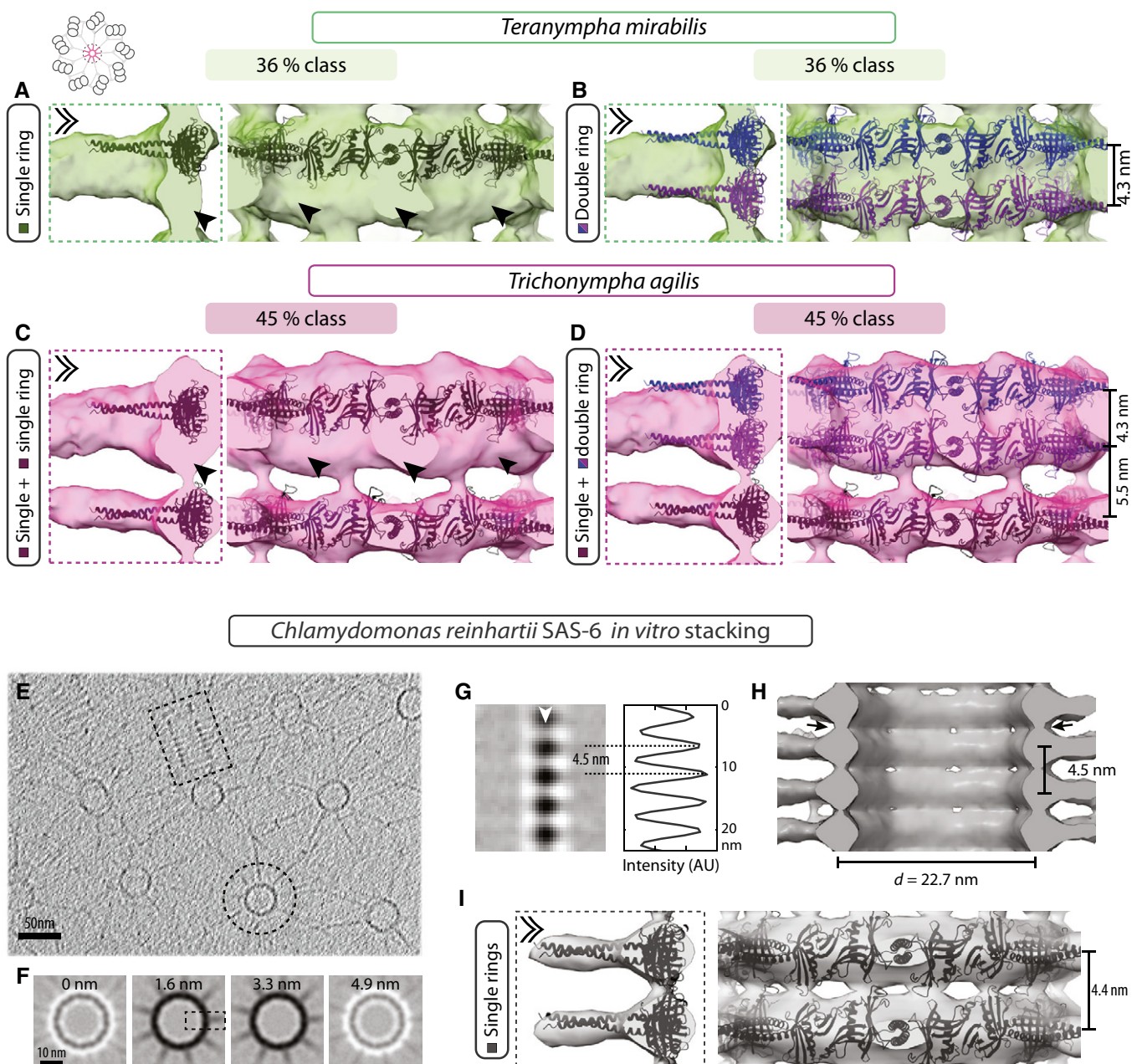

**Figure 5. Direct stacking of SAS-6 rings.**

A–D Computationally assembled CrSAS-6[6HR] single (A, C) or double ring in register (ribbon diagram shown in different shades for clarity) (B, D) fitted into the 3D maps of the 36% *T. mirabilis* class (A, B) and the 45% *T. agilis* class (C, D). Dashed box shows longitudinal section through hub element (left), double chevron viewing point of longitudinal external views (right). Extra unaccounted densities in single ring fitting are indicated by arrowheads (A, C). Note that each elongated hub density can accommodate two tightly stacked CrSAS-6[6HR] rings (B, D), and that only one ring can fit into the thin hub element in the 45% *T. agilis* class (C, D). Indicated distances stem from measurements on the fitted models.

E   Purified CrSAS-6[NL] protein self-organized into stacks and analyzed by cryo-ET; one longitudinal view and one transverse view are highlighted by a dashed rectangle and a dashed circle, respectively.

F, G   2D views through STA of *in vitro* self-assembled CrSAS-6[NL] proteins. Transverse sections at the indicated heights through one assembly unit (F), as well as longitudinal view (G) of the area delineated by a dashed box in (F); white arrowhead denotes position of line scan along the vertical axis at the level of the hub, with corresponding pixel intensities in arbitrary units (AU). The average distance between two hub elements is 4.5 ± 0.3 nm (N = 7; dashed lines).

H   Longitudinal view of *in vitro* self-assembled CrSAS-6[NL] proteins STA 3D map. Note densities bridging successive hubs vertically (arrows).

I   Two CrSAS-6[HR] single rings fitted into the 3D map of *in vitro* self-assembled CrSAS-6[NL] proteins. Dashed box indicates longitudinal section through hub element (left), double chevron indicates viewing point of longitudinal external view (right). Note that the 3D map can accommodate rings stacked 4.4 nm apart (measured on fitted ring models) and that vertical densities linking hubs are partially occupied.

fCID, which runs throughout the center of the cartwheel-bearing portion of the *T. mirabilis* centriole. Although the fCID seems disconnected from the hub, small elements linking the two can be discerned in the *T. mirabilis* 36% class (Fig 6D, see Fig EV2N and P, arrowheads). Moreover, the fCID might be linked with the hub through other protein segments that are small, flexible, or exhibit different periodicities, rendering them difficult to detect in the current STA map. It will be interesting also to address whether the fCID exhibits helical features. Elements related to the fCID may be present in systems other than *T. mirabilis*. Thus, electron-dense material is discernable in the geometrical center of the hub in *T.* spp. and less so in *T. agilis* (see Fig 2D and I), although it is discontinuous and smaller in diameter than the fCID. Moreover, a density that may be related to either CID or fCID is apparent inside the hub in other species, including *Chlamydomonas, Chytrid* and several insects (Olson & Fuller, 1968; McNitt, 1974; Geimer & Melkonian, 2004; O'Toole & Dutcher, 2014; Gottardo et al, 2015; Guichard et al, 2017; Uzbekov et al, 2018; Klena et al, 2020) (Fig 6A). Determining the molecular nature of CID and fCID in diverse systems is expected to help assess whether they share an evolutionary history and may serve a similar function.

We found conserved features also in the peripheral elements of the cartwheel-bearing region of the centriole. Thus, the pinhead is present in all three organisms analyzed here, as is the A-C linker, although the exact arrangement of moieties within these elements differs slightly. Together with the peripheral STA map in the proximal region of the *Chlamydomonas* centriole (Li et al, 2019), these findings indicate that peripheral elements have been significantly conserved across evolution. Slight differences between systems are observed notably in the microtubule triplets and prominent MIPs. At the resolution achieved here, we thus identified a prominent MIP in both *Trichonympha* species next to protofilament A9, close to protofilament A10, potentially corresponding to the A-microtubule seam (Ichikawa et al, 2017, 2019). A prominent MIP has also been observed at this position in centrioles of mammalian CHO cells and *Drosophila* S2 cells (Greenan et al, 2018, 2020; Li et al, 2019). The molecular identity of several *Chlamydomonas* MIPs present in ciliary axonemal microtubules was identified recently (Ma et al, 2019), but corresponding MIPs could not be detected here using the selected threshold, perhaps because not all periodicities were explored. Nevertheless, we found one additional prominent MIP next to protofilament A5 in *T.* spp., which appears to be conserved in metazoan centrioles (Greenan et al, 2018), and was identified as RIB72 and FAP252 in ciliary axonemal microtubules (Ma et al, 2019). Given the recent spectacular progress in assigning the molecular nature of many ciliary MIPs using high-resolution cryo-EM (Ma et al, 2019; Song et al, 2020), it is likely that better resolution STA maps combined with a search for different periodicities will also enable the efficient identification of centriolar MIPs.

### Root of proximal–distal polarity

The centriole is polarized along its proximal–distal axis, as evidenced for instance by the presence of the cartwheel and microtubule triplets in the proximal region versus microtubule doublets in the distal region. Likewise, distal and sub-distal appendages are restricted to the distal region, by definition. Ultimately, proximal–distal polarity of the centriole also translates into that of the axoneme that emanates from it. Prior work in *T.* spp. raised the possibility that centriole polarity might stem from the pinhead or the A-C linker, because both elements exhibit a clear proximal–distal polarity (Guichard et al, 2013). Whereas it remains plausible that polarity is imparted by peripheral elements in some settings, in light of the present findings an alternative view can be envisaged. Indeed, the offset between double SAS-6 rings generates an inherently asymmetric structure, which correlates with the polarized location of the CID, as well as with the asymmetric tilt angle of spoke densities that is also observed in distantly related eukaryotic groups (see Fig EV2 and Klena et al, 2020). Given that central elements are present before peripheral ones during organelle assembly in the canonical centriole duplication cycle, it is tempting to speculate that the asymmetric properties of offset double rings and/or the CID are key for imparting organelle polarity. The situation might differ during *de novo* centriole assembly, which in human cells does not require interactions between HsSAS-6 head domains (Wang et al, 2015). In this case, peripheral elements are thought to be more critical, and this may also be the case with respect to imparting organelle polarity.

### Working model of SAS-6 ring stacking

What are the mechanisms of SAS-6 ring stacking? The previous lower resolution *T.* spp. map (Guichard et al, 2013), together with an initial analysis of *in vitro* generated stacks of CrSAS-6[NL] (Guichard et al, 2017), led to the suggestion that the periodicity between subsequent SAS-6 ring oligomers is ~8.5 nm. The higher resolution analysis performed here in three species leads us to propose instead that SAS-6 rings stack directly on top of one another, thus doubling the number of SAS-6 molecules per vertical unit length. Although we cannot formally exclude that a single SAS-6 ring is present and that additional densities correspond to other unassigned protein(s), several pieces of evidence support the direct SAS-6 stacking scenario. First, two SAS-6 rings fit without clashes within vertically elongated hub densities in *Trichonympha* and *T. mirabilis*. Second, the *T. agilis* 45% class contains thin hub densities that can accommodate only one SAS-6 ring, reinforcing the notion that the thicker ones harbor two. Third, stacks reconstituted in a cell-free assay from purified CrSAS-6 exhibit direct ring superimposition.

Our findings taken together lead us to propose the following working model of SAS-6 stacking in the cellular context (Fig 6E and F). This model entails two modes of ring stacking. The first corresponds to the offset double ring configuration, with a tight packing of SAS-6 rings, which are ~3.2 nm apart for instance in the *T. mirabilis* 36% class. In line with our findings, directly superimposed hub densities with remarkably similar spacing are observed also in *Paramecium* and *C. reinhardtii* (Klena et al, 2020). In the second stacking mode, two such offset double rings stack in register, being ~5.2 nm apart from each other in the *T. mirabilis* 36% class. A similar alternating pattern is observed in *T.* spp. and *T. agilis*, albeit with slightly different distances, potentially owing to limitations in measurement precision. Overall, we propose that these two alternating modes of SAS-6 ring stacking represent a fundamental feature of cartwheel assembly.

The periodicity of the basic central cartwheel unit in this working model is ~16.8 nm, which corresponds to the largest conserved periodicity detected by FFT. This value is consistent with the distance

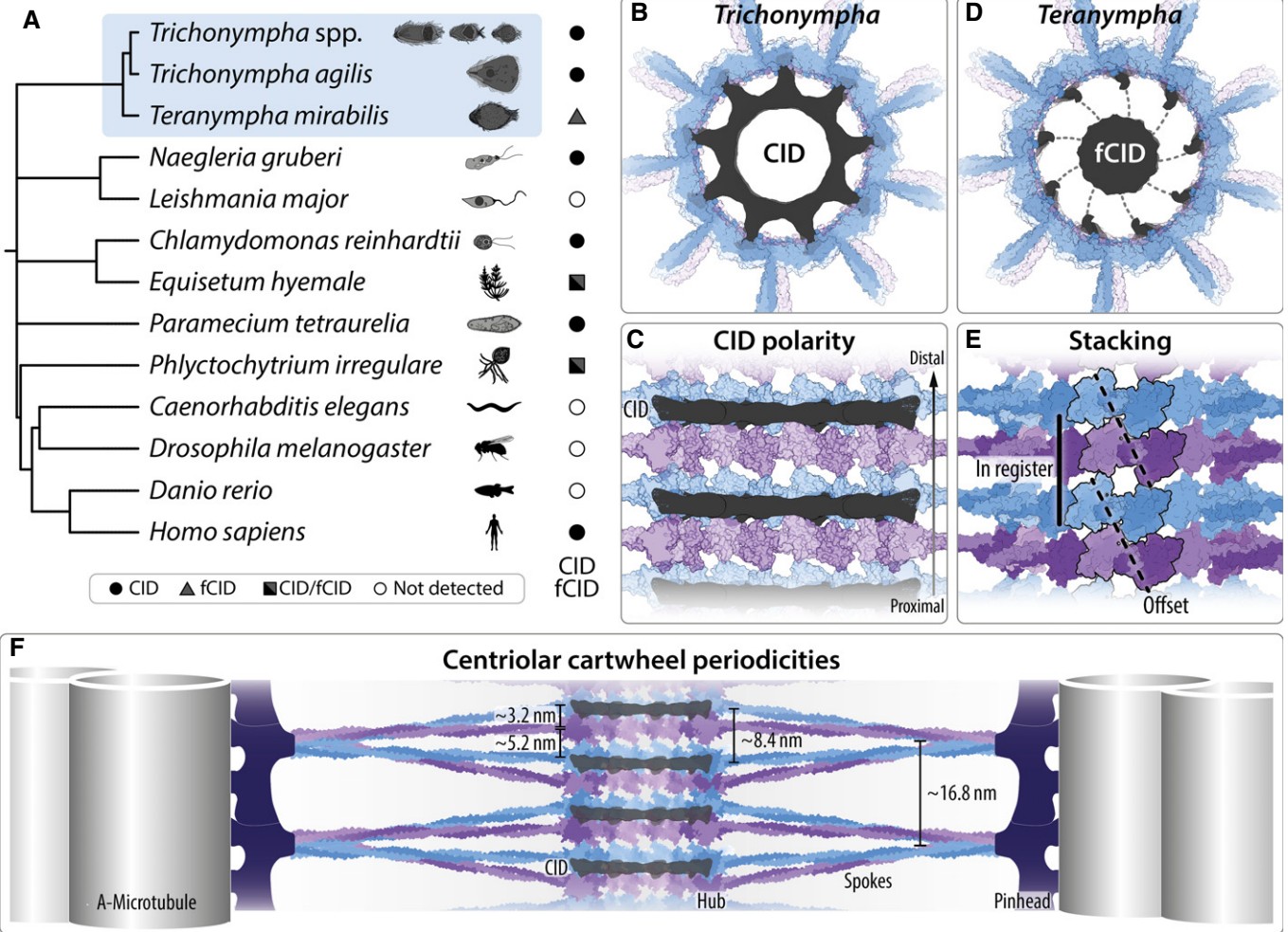

**Figure 6. Working model of centriolar cartwheel architecture.**

A   Occurrence of CID or fCID in species analyzed in this study (*T.* spp., *T. agilis*, and *T. mirabilis*, blue background), and in select other organisms (not to scale), mapped on a taxonomic tree generated using phyloTv2 based on NCBI taxonomy—https://phylot.biobyte.de/. See Discussion in main text for references.

B   Schematic of cartwheel hub in *T.* spp. and *T. agilis* with 9-fold symmetrical CID (gray) connecting to the hub, equidistant between two spokes.

C   The CID (dark gray) in *T.* spp. and *T. agilis* is polarized distally relative to the hub element comprising two tightly superimposed ring elements—shown in different colors for clarity.

D   Schematic of cartwheel hub in *T. mirabilis* with continuous fCID (dark gray) along the proximal–distal axis, with potential thin connections to the hub, equidistant between two spokes.

E   Working model of cartwheel stacking, with alternation of two modes of SAS-6 ring stacking: Pairs of rings are tightly superimposed with a rotational offset (dashed line), and such offset double rings are then stacked in register (solid line); the offset direction is consistent in *T.* spp., *T. agilis*, and *T. mirabilis*, yet may be different in other species, as shown in *Paramecium* (Klena *et al*, 2020). Surface representation of SAS-6 rings in different colors for clarity; homodimers highlighted by black contour.

F   Consensus model of periodicities in centriolar cartwheel along the proximal–distal axis. Surface representation of offset double rings tightly superimposed with ~3.2 nm periodicity, in register with the next such offset double ring located ~5.2 nm away, thus amounting to an overall periodicity of ~8.4 nm. Note that these values are based on the *T. mirabilis* 36% class, which has the most clearly resolved periodicities, yet does not have a CID as shown in this composite model, with related values in other species; see text for details. The CID also exhibits ~8.4 nm mean periodicity in the two *Trichonympha* species. Two spoke periodicities are observed: close to the hub, pairs of spokes are ~8.4 nm apart, whereas spoke pairs merge toward the periphery, where their periodicity is hence ~16.8 nm, matching that of the pinhead and connected microtubules. Note that spoke angles are merely illustrative and limited by the resolution of the available STA maps. Note also that alternative modes of spoke merging may be revealed by analyzing much larger sub-volumes and may be present in other species (Klena *et al*, 2020).

expected from an alternation of the two modes of SAS-6 stacking (2x ~3.2 nm for the double rings with an offset + 2x ~5.2 nm for such pairs of rings in register in the case of the *T. mirabilis* 36% class = ~16.8 nm) (Fig 6F). Each SAS-6 hub element possesses one spoke emerging from it, but owing to insufficient resolution, two closely superimposed spokes appear as one joint density in all cases, except for the single hub element in the *T. agilis* 45% class. Interestingly, we note also that hub elements with accompanying spoke densities are further apart in the *T. mirabilis* 64% class, indicative of a potential variation in stacking mode. Perhaps the densities

bridging successive hub elements in this case correspond to another protein than SAS-6. One intriguing candidate to consider in this context is SAS-6-like, an evolutionary ancient SAS-6 related protein lacking the coiled-coil (de Leon *et al*, 2013).

## Concluding remarks

In summary, conducting cryo-ET in three flagellates enabled us to uncover important conservation of centriolar architecture, as well as some species specificity. In the future, higher resolution analyses, including without symmetrizing, is expected to reveal additional polarity and stacking features, as well as to enable one to visualize protein domains and thus further unravel the mechanisms governing centriole assembly.

# Materials and Methods

### Transmission electron microscopy of *Trichonympha and Teranympha* cells

*T. agilis* and *T. mirabilis* were extracted from the hindgut of *Reticulitermes speratus* termites (a kind gift of Yuichi Hongoh, Tokyo University of Technology, Japan), as described (Guichard *et al*, 2015), placed in a drop of 10 mM K-PIPES, and gravity-sedimented for 3–5 min onto coverslips coated with poly-D lysine (Sigma-Aldrich, catalog number P1024), until cells came in contact with the glass. Cells were fixed overnight in 2% paraformaldehyde 1% glutaraldehyde in phosphate buffer 0.1 M, pH 7.4 (with 0.2% Tween-20 for some samples), washed in cacodylate buffer (0.1 M, pH 7.4) at 4°C, and post-fixed with 0.5% tannic acid in cacodylate buffer (0.1 M, pH 7.4) for 40 min at room temperature. After two washes in distilled water, cells were post-fixed in 1% osmium tetroxide in cacodylate buffer. Thereafter, samples were washed in distilled water and dehydrated in a graded ethanol series (1× 50%, 1× 70%, 2× 96%, 2× 100%). Finally, samples were embedded in epon resin and polymerized at 65°C overnight. 50 nm sections were cut and then stained in 1% uranyl acetate in distilled water for 10 min, followed by Reynold's stain (1.33 g lead citrate, 1.76 g sodium citrate, 160 mM NaOH in 50 ml distilled water (Reynolds, 1963)) for 5 min. Images were acquired using Tecnai Spirit at 80 kV or Tecnai F20 at 200 kV microscopes (Thermo Fischer Scientific). Pear-shaped *T. agilis* cells with long flagella covering the anterior of the cell can be readily distinguished from elongated *T. mirabilis* cells with shorter flagella arranged in rows spiraling around the cell. These distinct morphologies are preserved in resin. Purified centrioles from the two species were distinguished in cryo-ET based on the presence or absence of the fCID. Images of centriole cross-sections from TEM and cryo-ET were circularized and symmetrized using the CentrioleJ plugin (Guichard *et al*, 2013).

### Purification of *Trichonympha* and *Teranympha* centrioles

*Trichonympha* and *Teranympha* cells were extracted from the hindgut of termites in 10 mM K-PIPES in the presence of cOmplete protease inhibitor cocktail (1:1,000; Roche) as described (Guichard *et al*, 2015), with minor species-specific modifications. For the purification of centrioles from mixed *T. agilis* and *T. mirabilis*

populations from *R. speratus* termites, cells were pelleted at 1,000 *g* for 20 s and the supernatant discarded. For enrichment of *T. mirabilis* centrioles, cells were sedimented on ice 3 × 10 min in 1 ml 10 mM K-PIPES; the more fragile *T. agilis* cells spontaneously lysed during this step and thus were depleted from the resulting pellet. *T.* spp. (*T. collaris*, *T. campanula,* and *T. sphaerica*) were extracted from *Zootermopsis angusticollis* termites (a kind gift from Filip Husnik and Patrick Keeling, University of British Columbia, Vancouver, Canada), which harbor the same set of *Trichonympha* species as *Zootermopsis nevadensis* (Tai *et al*, 2013; Boscaro *et al*, 2017). After extraction, *T.* spp. cells were sedimented on ice 3 × 10 min in 1 ml 10 mM K-PIPES. In all cases, centrioles were released during a 20–40 min incubation on ice in 1 ml K-PIPES 10 mM 0.5% NP-40 plus protease inhibitors. Centrioles and associated flagella were either pelleted at 500 g for 3 min, or the resulting supernatant spun at 1,000 *g* for 5 min at 4°C; both types of preparations were utilized for cryo-ET. In all cases, centrioles were washed once in 1 ml 10 mM K-PIPES plus protease inhibitors and pelleted at 1,000 *g* for 5 min. The pellet was stored on ice before preparing grids for cryo-ET.

### Cryo-ET data acquisition

Grids were prepared as described previously (Guichard *et al*, 2015). Briefly, 4 μl of the purified centriole solution was mixed with 10 nm gold beads using a 20 μl tip cut at the extremity and then applied to the copper side of a glow-discharged lacey holey carbon grid (carbon Cu 300 mesh, Ted Pella). The grid was then manually blotted from the carbon side with a Whatman grade 1 filter paper and plunged into liquid ethane at −180°C with a manual plunge freezer.

Tilt series of centrioles that were approximately parallel to the tilt axis were collected with the Tomo 4.0 software (Thermo Fisher Scientific) on a Tecnai F20 TEM operated at 200 kV (Thermo Fisher Scientific); data were recorded with a Falcon III DD camera (Thermo Fisher Scientific) in linear mode at 29'000× magnification, corresponding to a pixel size of 3.49 Å. The tilt series were recorded from −60° to 60° using both continuous and bidirectional tilt schemes, with an increment of 2° at 2.5–5 μm underfocus. To allow determination of the proximal–distal axis, the blunt proximal end was included in image acquisition. A total of 54 tilt series were collected (Appendix Fig S1 and Appendix Table S1).

Tilt series alignments using gold fiducials were performed with IMOD v4.9 (Kremer *et al*, 1996). The contrast transfer function (CTF) was estimated with CTFFIND4 v1.8 (Rohou & Grigorieff, 2015) using relion_prepare_subtomograms.py (Bharat & Scheres, 2016). Variable SkipCTFCorrection was set to "True" to generate a wedge-shaped tilt- and dose-weighted 3D CTF model. Aligned tilt series were corrected by phase-flipping with ctfphaseflip and used for weighted back-projection (WBP) tomogram reconstruction with IMOD.

### Sub-tomogram processing

Areas corresponding to the central cartwheel region (CW) and to the peripheral microtubule triplet region (MTT) were identified visually in individual tomograms and their longitudinal axes modeled with closed contours in 3dmod. Proximal–distal polarity of the centriole was assessed by the fact that the proximal end was blunt,

in contrast to the distal end, where the flagellum was usually present. Moreover, the clockwise chirality of the microtubule triplets when viewed from the distal end provided an independent means to determine proximal–distal polarity, which was registered and maintained in the resulting sub-volumes. Also, reconstructions from sub-volumes re-centered on the spokes with CW and polar CID density on the one side, and spokes and polar pinhead on the other, allowed unambiguous polarity assignment (see Fig EV2).

Individual model points were interpolated along the contours using the addModPts from the PEET package (Heumann *et al*, 2011), with a step size of 3 × 85 Å = 252 Å corresponding to a 73 pixels shift (see Appendix Fig S1B). The interpolation step size was selected to correspond to approximately three hub elements, as smaller step sizes could not separate cartwheel variations visible in the raw tomograms (see Appendix Fig S1D). Larger step sizes were also explored to generate non-overlapping sub-volumes along the proximal–distal axis of the cartwheel hub, shifted relative to each other by more than half of the sub-volume size. The resulting STAs exhibited similar structural features, yet were noisier due to the reduced number of sub-volumes.

Interpolation with a step size of 252 Å resulted in 1,385 initial CW sub-volumes for *T.* spp., 1,154 for *T. agilis* and 1,802 for *T. mirabilis*. A similar approach and step size were selected for extraction of MTT, resulting in 2,792 initial MTT sub-volumes for *T.* spp., 3,850 for *T. agilis* and 7,928 for *T. mirabilis* (see Appendix Fig S1G–I; Appendix Table S1). All further processing was performed with RELION v2.1 or v3.1 (Kimanius *et al*, 2016). Briefly, 2D projections of all sub-volumes along the Z axis were generated and subjected to reference-free 2D classification. 2D class averages with clear CW or MTT densities were selected for further data processing. At this step, the hub of both *T. agilis* and *T. mirabilis* showed variations in vertical spoke and hub element spacing. Power spectra of 2D class averages were used to measure the vertical spacing of the CW elements (see Appendix Fig S2C, Appendix Fig S3C and F, Appendix Fig S4C and F) or MTT (see Appendix Fig S2F, Appendix Fig S3I, Appendix Fig S4I). Modeling with 3dmod was also used to investigate the proximal–distal twist of microtubule blades. No twist angle of the microtubules was observed along the proximal–distal axis of the cartwheel-bearing region in any of the species, as previously reported for *T.* spp. (Gibbons & Grimstone, 1960).

Selected 2D projections of CW sub-volumes were re-extracted, now as 3D sub-volumes with a box size of 200 pixels (corresponding to 80 nm, with the exception of *T.* spp., which corresponds to 88 nm) and re-scaled to a pixel size of 4 Å. Initial 3D CW and MTT references were built *de novo* with the Stochastic Gradient Descent (SGD) algorithm from the re-extracted 3D sub-volumes. The resulting CW initial reference was re-aligned with relion_align_symmetry to orient the visible nine-fold rotational (C9) axis along the Z axis, and C9 symmetry applied. Similarly, the MTT initial reference proximal–distal axis was re-aligned with the Z axis. Consensus 3D refinement of the CW initial reference with C9 symmetry and missing wedge correction revealed a well-resolved CW density. Consensus 3D refinement was followed by focused 3D refinement with a soft mask that included only the CW hub with the CID, as well as part of the emanating spoke density, whose inclusion inside the mask was important to avoid misalignment along the proximal–distal axis. Moreover, one round of focused 3D classification in *T. agilis* and

*T. mirabilis* with a soft tight mask that included only three central stacked rings revealed vertical spacing variations in 3D.

The *T. agilis* CW hub was classified into two groups representing 55 and 45% of refined sub-volumes. The 55% class showed uniform 8.5 nm vertical spacing measured from the 3D reconstruction in real space and from the power spectrum. Further 3D classification did not yield any observable improvement in the quality of the maps as judged by features and resolution. In contrast, further focused 3D classification without alignment of the 45% class revealed two subgroups corresponding to 20 and 25% of all refined sub-volumes, with differences in distances between alternating thin and thick rings. The *T. mirabilis* CW hub was classified into two main groups representing 64 and 36% of refined sub-volumes. The 64% class exhibited ∼8.4 nm vertical spacing measured from 3D reconstruction in real space, and of 8.5 nm from power spectrum; this class also exhibited a smooth CW hub surface. In contrast, the 36% class revealed ∼8.4 nm vertical spacing with two smaller units per hub with 3.2 nm and 5.2 nm spacing measured from 3D reconstruction in real space, and 4.3 nm from power spectrum. Further 3D classification did not yield any observable improvement in the quality of the maps as judged by features and resolution. For all 3D classes, final focused 3D refinements with local alignments were performed. Analogous analyses of *T.* spp. revealed a single stable class with uniform 8.5 nm vertical spacing measured from the 3D reconstruction and from the power spectrum.

To establish the polar relationship between the CID, the corresponding CW, and the peripheral elements (A-microtubule, pinhead, A-C linker), the symmetry was relaxed from C9 to C1 with relion_particle_symmetry_expand. The resulting symmetry-expanded sub-volumes were re-extracted with binning 3 and a box size of 100 pixels, using the "re-center refined coordinates" option approximately on the spokes center to include the CW density on one side and spokes and pinhead on the other. Focused 3D classification of re-extracted sectors without alignment and without symmetry iterated with focused 3D refinement revealed reconstructions with well-resolved connections between central and peripheral elements. No structural features with a periodicity different than a multiple of 8.5 nm were detected at this step.

A similar processing pipeline was applied to peripheral elements (microtubules, pinhead, A-C linker). Selected 2D projections of MTT sub-volumes were re-extracted as binned 2 times 3D sub-volumes with a box size of 128 pixels corresponding to 89 nm to include microtubule triplet and emanating pinhead and A-C linker densities (see Appendix Fig S1B). Initial 3D MTT references were built *de novo* with the SGD algorithm from the re-extracted 3D sub-volumes. The resulting MTT initial reference proximal–distal axis was re-aligned with the Z axis. Re-extracted MTT sub-volumes were subjected to consensus 3D refinement followed by rounds of focused 3D classification and 3D refinement. Refined sub-volumes were re-extracted either at the center of emanating pinhead or A-C linker densities and refined separately. Local resolution distributions were determined with ResMap (Kucukelbir *et al*, 2014), whereas directional 3D FSC were measured with 3DFSC web server (Zi Tan *et al*, 2017).

### Rigid-body fitting and computational assembly of SAS-6 rings

Rigid-body fitting of CrSAS-6[6HR] (PDB-3Q0X) into *T.* spp., *T. agilis* and *T. mirabilis* 3D STA maps was performed with

ADP_EM plugin (Garzón *et al*, 2007), followed by symmetrical fitting in UCSF Chimera (Pettersen *et al*, 2004).

Single CrSAS-6[6HR] rings were computationally assembled as described (Kitagawa *et al*, 2011), while double rings were assembled by superposing two such CrSAS-6[6HR] rings using the crystallized stacks of LmSAS-6 rings as guide (van Breugel *et al*, 2011). Offset double rings were assembled manually by rotating the proximal ring clockwise, with the angle being estimated based on offset between fitted homodimers. The vertical distance between rings in the double offset ring configuration was reduced by 0.4 nm, and clashes were only allowed between flexible loops. CrSAS-6[6HR] single ring, double in register rings, or double offset rings were fit in the map using *fitmap* function of ChimeraX (Goddard *et al*, 2018). Coordinate models were initially converted to molecular maps at corresponding global resolutions and then fitted into the STA maps. Calculated values for correlation and overlap between maps are reported in Appendix Table S2. UCSF Chimera or ChimeraX were used for visualization and segmentation of 3D models.

### *In vitro* assembly of CrSAS-6 stacked cartwheels

A 6xHis- and S-tag-containing CrSAS-6 construct harboring the N-terminal globular head domain plus the entire coiled-coil (referred to a CrSAS-6[NL], comprising amino acids 1–503) was expressed and purified as described (Guichard *et al*, 2017), except that the tag moieties were removed by TEV protease cleavage. The resulting purified CrSAS-6[NL] protein was dialyzed overnight from Tris pH 7.5 150 mM NaCl into 10 mM K-PIPES pH 7.2 using a mini dialysis unit at 4°C (slide-A-lyzer, 3.5 K, 10–100 ml, Pierce, catalogue number 69550). Thereafter, 5 μl of dialyzed material was pipetted onto a glow-discharged Lacey holey carbon grid (Carbon Cu 300 mesh, Ted Pella), blotted for 3 s (blot force −15, no wait time), and plunge-frozen by vitrification in liquid nitrogen-cooled liquid ethane, using a Vitrobot MKIV (Thermo Fisher Scientific).

Tilt series were collected on a Titan Krios TEM (Thermo Fisher Scientific) operated at 300 kV and equipped with a Gatan Quantum LS energy filter (zero loss slit width 20 eV; Gatan Inc.) on a K2 Summit direct electron detector (Gatan Inc.) in electron counting mode (110 $e^-$/Å$^2$ total dose), at a calibrated pixel size of 2.71 Å. The tilt series were recorded from −60° to 60° using a dose-symmetric tilt scheme (Hagen *et al*, 2017) with increments of 2° at 2.5–3.5 μm defocus. A total of 31 tilt series were collected automatically using SerialEM (Mastronarde, 2005).

Tilt series alignment using gold fiducials was performed with IMOD v4.9. The CTF was estimated with CTFFIND4 v1.8 using relion_prepare_subtomograms.py. Variable SkipCTFCorrection was set to "True" to generate a wedge-shaped tilt- and dose-weighted 3D CTF model. Aligned tilt series were corrected by phase-flipping with ctfphaseflip and used for WBP tomograms reconstruction with IMOD. Further STA was performed as described above for *Trichonympha* centrioles. Briefly, side views of CrSAS-6[NL] stacks were identified visually in tomograms and their longitudinal axes modeled with closed contours in 3dmod. Next, individual model points were added along the contours using addModPts with a step size of 2 CrSAS-6[NL] rings (2 × 42 Å = 84 Å) to prevent misalignment of potentially merged spokes, resulting in 926 initial sub-volumes (see Appendix Fig S1J;

Appendix Table S1). Further processing was performed with RELION v2.1 or v3.1 as described above.

## Data availability

Sub-tomogram averages have been deposited at the Electron Microscopy Data Bank (https://www.ebi.ac.uk/pdbe/emdb/) with accession codes EMD-10916, EMD-10918, EMD-10922, EMD-10923, EMD-10927, EMD-10928, EMD-10931, EMD-10932, EMD-10934, EMD-10935, EMD-10937, EMD-10938, EMD-10939, EMD-10941, and EMD-10942 (see Appendix Table S1).

**Expanded View** for this article is available online.

## Acknowledgements

We are grateful to Yuichi Hongoh, Patrick Keeling, and Filip Husnik for providing termites. We thank Graham Knott and Stéphanie Rosset (BioEM platform of the School of Life Sciences, EPFL) for assistance with TEM, as well as Kenneth Goddie, Lubomir Kovacik, and Henning Stahlberg (Center of Nanoimaging, Biozentrum, Basel, Switzerland) with data collection on the Titan Krios. Niccolò Banterle, Graham Knott, and Fabian Schneider are acknowledged for their critical reading of the manuscript. This work was funded by grant from the European Research Council to PG (AdG 340227) and the Swiss National Science Foundation (SNSF) to PaG (PP00P3_157517).

## Author contributions

SN, AB, GNH, VNV, and DD collected and analyzed data. SN, AB, GNH, and PGo contributed to experimental design and analysis, as well as writing the manuscript. PGu and MLG contributed to data analysis. All authors read and approved the final manuscript.

## Conflict of interest

The authors declare that they have no conflict of interest.

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
