## [Review Process File · The EMBO Journal]

Novel features of centriole polarity and cartwheel stacking revealed by cryotomography

Sergey Nazarov, Alexandra Bezler, Georgios Hatzopoulos, Veronika Nemčíková Villímová, Davide Demurtas, Maeva Le Guennec, Paul Guichard, and Pierre Gönczy

DOI: [10.15252/embj.2020106249](https://doi.org/10.15252/embj.2020106249)

Corresponding author(s): Pierre Gönczy (Pierre.Gonczy@epfl.ch)

Review Timeline:	Transfer from Review Commons:	15th Jul 20
	Editorial Decision:	31st Jul 20
	Revision Received:	11th Aug 20
	Accepted:	21st Aug 20

Editor: Hartmut Vodermaier

Transaction Report: This manuscript was transferred to The EMBO Journal following peer review at Review Commons.

**Review
COMMONS**

Reviewer #1

Centriole structure has been an attractive but challenging research topic for years. Pierre Gönczy's group has been working on its structure using cryo-electron tomography (cryo-ET). While the axoneme, which has longitudinal periodicity, was analyzed by several groups by cryo-ET for more than a decade, cryo-ET study on the centriole suffers from poor signal to noise ratio due to its limited length and thus fewer periodicity. They chose the centriole of flagellate *Trichonympha*, which have exceptionally long centrioles and thus offer opportunity of relatively straightforward subtomogram averaging. Their approach has been successful and they revealed intermediate resolution structure of the cartwheel, key of 9-fold symmetry formation, and its joint to triplet microtubules (Guichard et al. 2012, 2013, 2018).

In this work, they employed modern state-of-art cryo-ET technique, such as direct electron detection and 3D image classification to upgrade our knowledge of centriole structure. In their past works, the central hub of the cartwheel, made of SAS-6 protein forming 9-fold complex, was described as an 8nm periodic object. With improved spatial resolution, they provided further detail with clear polarity, which will deepen our thought about the initial stage of ciliogenesis. They also compared two *Trichonympha* species (spp and *agilis*) as well as another flagellate, *Teranympha micabilis*, and extended their intriguing evolutionary and mechanical hypotheses based on structural differences.

Despite improved spatial resolution, it is still not possible to identify proteins in the cryo-ET map (cellular cryo-ET will not reach such high resolution in the near future). Therefore this work is rather geometrically descriptive, which will inspire molecular biologists to identify molecules by other methods. Nevertheless this work demonstrated capability of cellular cryo-ET, especially analysis of structural heterogeneity. Thus, while biological topics handled are rather specialized for cilia from flagellate, this work will attract attention of any biologist interested in molecular structure in vivo. It is worth for publication in a high Journal after addressing the points below. This reviewer believes that the authors can address these points easily with additional analysis.

Major points:

1. Entire scheme

A graphic diagram of the entire cartwheel area, summarizing this work, is necessary for the readers' understanding (similar to Fig.6 of the other manuscript, Klena et al.). Then average scheme should be shown in more detail, especially assumption of periodicity, Materials and Methods. The cartwheel hub was averaged with 25nm periodicity (as discussed below). Was the pinhead averaged with 16nm (as detected by FFT in Fig.S2L)? How about the triplet?

This reviewer is not completely sure if the longitudinal averaging strategy is justifiable. Since periodicity of each domain is not trivial, logically the initial average must be done with the size of least common multiple (or larger). It is likely 96nm, assuming 25nm of the central hub is 3 times of microtubule periodicity and 16nm of the pinhead is twice of MT. 96nm average should be possible with a long cartwheel in this work. Alternative, in case periodicity is independent of MT and thus there is no least common multiple, is random picking and classification mentioned in "4. Periodicity". This should also be possible, since they can pick enough number of particles from long cartwheels.

2. Classification

The authors analyzed structural heterogeneity inside the cartwheel hub, employing reference-free classification by Relion software. The program reveals multiple coexisting structures - two from *Trichonympha agilis* and three from *Teranympha*, respectively. Whereas this is an exciting finding and shows future research direction of this field, interpretation of this classification must be done carefully.

It is puzzling that major (55%) population of *T. agilis* shows more ambiguous features than the minor population (45%), while spatial resolutions by FSC are not so different - for example,

Fig.2H vs Fig.S5C. In case of *Teranympa*, it is even more drastic - Fig.4D (major class) seems blurred along the centriolar axis, compared to Fig. 4E (minor class). This reviewer is afraid that these "major" classes might contain more than one structure and after subaveraging be blurred in detailed features. The apparent good spatial resolution could be explained, when two structures coexist and subtomograms are aligned within each subclass. Probably lower resolution at the spoke region of the major class (Fig.S2A) than that of the minor class (Fig.S2D) is a sign of heterogeneity within this class. Another risk could be subtomograms with poorer S/N being categorized to one class (due to lack of feature to be properly classified). Fig.S5F (black dots localized in one tomogram) raised this concern.

The following investigation will help to solve this issue. 1. Extract and re-classify subtomograms belonging to the major population. 2. Direct observation of tomograms. The authors could plot two classes of *Teranympa* (as they did for *T. agilis* in Fig.S5) and find features of the cylindrical cartwheel hub in two conformations (as shown Fig.4DE). Since such a feature was directly observed in tomograms from the other manuscript (left panels of Fig.S6AC in Klena et al.), it should be possible in this work as well.

3. Periodicity mismatch

In Fig. 2CD, periodicity of CID has discrepancy from that of the stacked SAS-6 ring (8.5nm and 8.0nm). Do the authors think this is a significant difference or within an error? The same question can occur to other subtomogram averages. It would be nice to show errors as shown in their other manuscript (Fig.3C of Klena et al.) and clarify their idea. If it is systematic difference of periodicity between the stacked ring and CID, this shift will be accumulated through the entire cartwheel region - after 100nm, 8.5nm/8.0nm difference can be accumulated to ~6nm, which should change the entire view of the subtomogram - and the main factor to be classified (periodicity mismatch). This artifact (or influence) should be removed (or separately evaluated) by masking CID (out and in) and run classification separately. By clarifying this, the quality of the major subaverages (mentioned in the previous paragraph) could be improved.

4. Periodicity

They averaged subtomograms extracted with spacing of 252Å with initial average as the first template (p.18 Line22). This means they assumed 25nm periodicity from the beginning and excluded different or larger unit size (if they take search range wide, they could detect difference periodicity, but will still be biased by initially assumed 25nm). 25nm average allowed them to see more detail than before (when they assumed 8nm periodicity), but there is still a risk of bias from references. To avoid this risk, this reviewer would propose classification of randomly extracted (but of course along the cylindrical hub or along the triplet microtubules, so one-dimensionally random picking) subtomograms. This experiment will end up with multiple subaverages, which are 25nm (or multiple times of that) shifted from each other. Then it will prove their assumption.

Minor points:

They discussed difference of stacked SAS-6 rings in the cartwheel from various species. How much is the sequence difference of SAS-6 among these species?

Are the authors sure that CID is nine-fold symmetric? It is not trivial.

p.7 Line21 "Fig.S1D-O": D-L

p.8 Line1: It would be nice if more detailed description about MIPs, correlating to recent high resolution works from Bui and Brown labs.

p.9 Line6 "Focused 3D classification...": This sentence is unclear.

p.18 5 lines from bottom "S6C, S6F": How can these panels be power spectra to measure spacing? Typo?

Fig.1C: Another cross-section from the distal region will be helpful. A longer scale bar is better for readers' understanding.

p.29 Line6: pin -> pink

Fig.S6F: It would be informative if the subclasses (25% and 20%) are distinguished in this mapping.

A figure to explain the classification scheme will help readers understand. How many

subtomograms did classification started? Were the 45% class classified into two (25% and 20%) groups by two-step classification or at once (the entire subtomograms were classified into three groups directly)?

Significance

Nevertheless this work demonstrated capability of cellular cryo-ET, especially analysis of structural heterogeneity. Thus, while biological topics handled are rather specialized for cilia from flagellate, this work will attract attention of any biologist interested in molecular structure in vivo. It is worth for publication in a high journal after addressing the points above. This reviewer believes that the authors can address these points easily with additional analysis.

Reviewer #2

Here, Nazarov and colleagues report sub-tomogram average (STA) maps of centrioles with 16 to 40 Å resolution from *Trichonympha* spp., *Trichonympha agilis*, and *Teranympha mirabilis*. Even though the authors have previously described the centriole architecture of *T. spp.*, these STA maps of higher resolution revealed new features of centrioles, like polarized Cartwheel Inner Density (CID) and the pinhead. They also observed Filament-like structure (FLS) from *T. mirabilis* which seems to correspond to the CID from other species. Interestingly, they suggest that one and two SASS6 rings are stacked in an alternative fashion to make the central hub in *T. m. mirabilis* (Figure 5). The following issue should be addressed:

Major points

1. Figure 4E. Authors mentioned in the manuscript that "We observed that every other double hub units in the 36% *T. mirabilis* class appears to exhibit a slight tilt angle relative to the vertical axis". When I see the other side, it does not seem to be tilted. Could the authors explain this?

Minor Points

1. Page 11, I think Fig. 9G indicates Fig. S9G.

Significance

I believe these results are of interest for all centrosome researchers, and would like to recommend this manuscript be published in the EMBO journal which is affiliated with the Review Commons.

Reviewer #3

In this manuscript Nazrov et al., use cryo-electron tomography (CET) to analyse the structure of the centriole cartwheel. The Gonczy lab have previously generated a ground-breaking structure of the cartwheel from *Trichonympha* spp (*T. spp.*) (Guichard et al., *Science*, 2012; Guichard et al., *Curr. Biol.*, 2013). This work is a direct continuation of those studies but using modern technology to get higher resolution images of the *T. spp.* cartwheel, and comparing this to the cartwheel from *Trichonympha agilis* and from another distantly related flagellate *Tetranympha mirabilis*.

The data is generally well presented and of high quality. I am not an expert in CET, so it would be advisable to get the opinion from a reviewer who is, but the Gonczy lab are experienced in these techniques so I would not anticipate any problems. I have to admit that the title of the paper did not excite me, and I expected this to be a very worthy, but incremental study. It was a

pleasure to find out that the extra detail provided by the increased resolution has revealed several new and unexpected features that have important implications for our understanding of cartwheel assembly and function. Most important are the potential asymmetry of the cartwheel hub, apparent variations in the packing mechanism of the stacked rings (even within the same cartwheel), and the potential offsetting of ring stacking. These findings will be of great interest to the field, and so I am strongly supportive of publication in The EMBO Journal. I have only a few points that I think the authors should consider.

1. Nazarov et al., conclude that the cartwheel structure is intrinsically asymmetric. This is most convincingly based on the displacement of the CID within the hub, but they state that the Discussion that the potential offset between the Sas-6 double rings generates an inherently polar structure. I didn't understand why this is the case. Looking at Fig.S9A,B I can see that the offset in B could tilt to the left (as shown here) or to the right (if the structure was flipped by 180°). But I couldn't see how this makes this structure polar in the sense that a molecule coming into dock with the structure could only bind to one side of the offset structure shown in B, but to both sides of the aligned structure shown in A. I think this needs to be explained better, as it is crucial to understand where any potential polarity in the cartwheel structure comes from.

2. Related to this last point, in a co-submitted paper Klena et al. do not report such an asymmetry in the hub structures they have solved from several different species (neither in the tilting of the hub, or the displacement of the CID). I think it would be worth both sets of authors commenting on this point.

3. The authors data strongly suggests that the *T. agg.* and *Te. mir.* hubs are composed of a mixture of single and double Sas-6 rings. In contrast, the *T. spp.* cartwheel only has a single class of rings, but it wasn't absolutely clear if the authors think this comprises a single or double ring. In the text it is presented as though the elongation of the hub densities in the vertical direction is a new feature of the *T. agg.* cartwheel (Fig.2H,I), but to me it looks as though this is also apparent in the *T. spp.* cartwheel (Fig.2C,D). The authors should address this directly and, if they believe that *T. spp.* has a double ring, they should comment on whether this more regular structure seems to have offset rings. If not, then the offset rings are unlikely to be the source of asymmetry that leads to the asymmetric displacement of the CID. Finally, if the authors think these are double rings, they should also be clear that they would now slightly re-interpret their original *T. spp.* cartwheel model (Figure 2, Guichard et al., *Curr. Biol.*). There is no embarrassment in this—a higher resolution structure has simply revealed more detail.

4. The authors conclude that *T. mirabilis* cartwheels lack a CID and instead have a filament-like structure (FLS). I wonder whether it is more likely that the FLS is really a highly derived CID that appears to be structurally distinct when analysed in this way, but that will ultimately have a similar molecular composition. This situation might be analogous to the central tube in *C. elegans*, which by EM appears to be distinct from the central cartwheel seen in most other species, but is of course still composed of Sas-6. This historical tube/cartwheel nomenclature is now cumbersome to deal with, so perhaps it would be better to be cautious and not give the *T. mirabilis* structure a completely new name—how about "unusual CID" (uCID).

Significance

See above.

We thank the reviewers for their useful suggestions to improve the manuscript and their support for publication. We have addressed all the comments that have been raised and carried out the suggested additional analyses, resulting in a significantly improved revised version of the manuscript. We provide hereafter a detailed point-by-point response to all questions and comments of the three reviewers.

Reviewer #1 (Evidence, reproducibility and clarity (Required)):

*Centriole structure has been an attractive but challenging research topic for years. Pierre Gonczy's group has been working on its structure using cryo-electron tomography (cryo-ET). While the axoneme, which has longitudinal periodicity, was analyzed by several groups by cryo-ET for more than a decade, cryo-ET study on the centriole suffers from poor signal to noise ratio due to its limited length and thus fewer periodicity. They chose the centriole of flagellate *Trichonympha*, which have exceptionally long centrioles and thus offer opportunity of relatively straightforward sub-tomogram averaging. Their approach has been successful, and they revealed intermediate resolution structure of the cartwheel, key of 9-fold symmetry formation, and it's joint to triplet microtubules (Guichard et al. 2012, 2013, 2018). In this work, they employed modern state-of-art cryo-ET technique, such as direct electron detection and 3D image classification to upgrade our knowledge of centriole structure. In their past works, the central hub of the cartwheel, made of SAS-6 protein forming 9-fold complex, was described as an 8nm periodic object. With improved spatial resolution, they provided further detail with clear polarity, which will deepen our thought about the initial stage of ciliogenesis. They also compared two *Trichonympha* species (*spp* and *agilis*) as well as another flagellate, *Teranympha mirabilis*, and extended their intriguing evolutionary and mechanical hypotheses based on structural differences. Despite improved spatial resolution, it is still not possible to identify proteins in the cryo-ET map (cellular cryo-ET will not reach such high resolution in the near future). Therefore, this work is rather geometrically descriptive, which will inspire molecular biologists to identify molecules by other methods. Nevertheless, this work demonstrated capability of cellular cryo-ET, especially analysis of structural heterogeneity. Thus, while biological topics handled are rather specialized for cilia from flagellate, this work will attract attention of any biologist interested in molecular structure in vivo. It is worth for publication in a high Journal after addressing the points below. This reviewer believes that the authors can address these points easily with additional analysis.*

We are grateful to the reviewer for the favorable evaluation and the many valuable suggestions, in particular concerning the processing pipeline, which we addressed by additional analyses, as detailed below.

Major points:

1. *Entire scheme*

A graphic diagram of the entire cartwheel area, summarizing this work, is necessary for the readers' understanding (similar to Fig.6 of the other manuscript, Klena et al.).

We thank the reviewer for this interesting suggestion, which we fully adhere to. As a result, we have generated a graphical summary of the work, which is shown in the new Figure panels 6B-F. Moreover, Figure 6A provides an evolutionary perspective regarding the presence of the CID and of what is now referred to as the fCID (filamentous CID, previously: FLS, see response to reviewer 3). This also helps to link our findings with the companion manuscript by Klena *et al.* This new Figure 6 is referred to extensively in the discussion of the revised manuscript (pages 13-16).

Then average scheme should be shown in more detail, especially assumption of periodicity, Materials and Methods. The cartwheel hub was averaged with 25nm periodicity (as discussed below). Was the pinhead averaged with 16nm (as detected by FFT in Fig.S2L)? How about the triplet?

This reviewer is not completely sure if the longitudinal averaging strategy is justifiable. Since periodicity of each domain is not trivial, logically the initial average must be done with the size of least common multiple (or larger). It is likely 96nm, assuming 25nm of the central hub is 3 times of microtubule periodicity and 16nm of the pinhead is twice of MT. 96nm average should be possible with a long cartwheel in this work. Alternative, in case periodicity is independent of MT and thus there is no least common multiple, is random picking and classification mentioned in "4. Periodicity". This should also be possible, since they can pick enough number of particles from long cartwheels.

We apologize that the initial version of the manuscript was not sufficiently clear regarding the averaging pipeline that was pursued. To rectify this, we now provide a new Figure S1B to graphically explain the approach followed for STA. As depicted in this figure panel, the step size for sub-volume extraction was 25 nm both centrally and peripherally. This step size was selected because it corresponds to ~3x the major periodicity of ~8.5 nm observed in the power spectra of the sub-volumes. The 25 nm step size is larger than that previously used (i.e. 17 nm in Guichard et al. 2013), in order to identify potential features with larger periodicities. The fact that the step size was of 25 nm in all cases is now mentioned explicitly in the Materials and Methods section of the revised manuscript (line 649).

We agree with the reviewer that 96 nm averaging is possible given the long cartwheel analyzed here, and such a piece of data was in fact included in the original submission, although with a different purpose. Indeed, we carried out STA using ~ $(100 \text{ nm})^3$ sub-volumes (with binning 3 to reduce computational time), the results of which are reported in Figure S7 (previously Fig. S6). For the purpose of this analysis, we focused on the lateral organization of the cartwheel, but did not use this dataset to explore other periodicities because of the limitations inherent to a binning 3 data set.

2. Classification

*The authors analyzed structural heterogeneity inside the cartwheel hub, employing reference-free classification by Relion software. The program reveals multiple coexisting structures - two from *Trichonympha agilis* and three from *Teranympha*, respectively. Whereas this is an exciting finding and shows future research direction of this field, interpretation of this classification must be done carefully. It is puzzling that major (55%) population of *T. agilis* shows more ambiguous features than the minor population (45%), while spatial resolutions by FSC are not so different - for example, Fig.2H vs Fig.S5C. In case of *Teranympha*, it is even more drastic - Fig.4D (major class) seems blurred along the centriolar axis, compared to Fig. 4E (minor class). This reviewer is afraid that these "major" classes might contain more than one structure and after subaveraging be blurred in detailed features. The apparent good spatial resolution could be explained, when two structures coexist and subtomograms are aligned within each subclass. Probably lower resolution at the spoke region of the major class (Fig.S2A) than that of the minor class (Fig.S2D) is a sign of heterogeneity within this class. Another risk could be subtomograms with poorer S/N being categorized to one class (due to lack of feature to be properly classified). Fig.S5F (black dots localized in one tomogram) raised this concern. The following investigation will help to solve this issue. 1. Extract and re-classify subtomograms belonging to the major population. 2. Direct observation of tomograms. The authors could plot two classes of *Teranympha* (as they did for *T. agilis* in Fig.S5) and find features of the cylindrical cartwheel hub in two conformations*

(as shown Fig.4DE). Since such a feature was directly observed in tomograms from the other manuscript (left panels of Fig.S6AC in Klena et al.), it should be possible in this work as well.

We agree with the reviewer that the interpretation of the classification must be done with care, and share her/his interest in better understanding the structural variability between cartwheels classes in *T. agilis* and *T. mirabilis*. Although poor S/N may in theory result in erroneous joint classifications, we note that all maps in the original submission stemmed from extensive focused 3D classification, which removed defective and spurious sub-volumes, nevertheless defining distinct classes in the cases reported. Obviously, however, we cannot exclude that much larger data sets and future software advances may lead to the identification of additional features that would allow further sub-classes to be identified.

Regardless, we followed the two suggestions the reviewer offered to us and have (1) extracted and re-classified sub-tomograms belonging to the major populations and (2) undertaken a direct observation of tomograms. These two points are developed in turn below.

(1) We have performed a further round of classification of the major populations in *T. agilis* (55 % class) and *T. mirabilis* (64 % class), to assess whether additional sub-classes might be identified and thus help further improve the quality of the central cartwheel map. However, this additional round did not yield new sub-classes nor notable improvement in the map quality as judged by visual inspections. We show in Rebuttal Figure 1 a comparison in each case of the original STA and the corresponding STA upon such re-classification. Importantly, all conclusions spelled out in the original submission hold upon further re-classification, indicating that the initial classification converged to the best map quality based on the current data set and available computational resources.

Rebuttal Figure 1: Re-classification of major classes

(A-D) Transverse (top) and longitudinal (bottom) views of *T. agilis* (A, B) and *T. mirabilis* (C, D) central cartwheel 3D maps. The final major classes reported in the manuscript (A: 55 % class, C: 64 % class) were subjected to re-classification, which again yielded one major class in each case, with no notable improvement (B, D).

(2) We have followed the suggestion of the reviewer and now show raw tomograms to confirm that the classes correspond to *bona fide* structures and not to processing

artefacts (new Figures S1C-F). The resulting new Figure S1D for instance shows that the striking variations observed between classes in the *T. agilis* STA are also visible in the raw tomogram. The more subtle variations among *T. mirabilis* classes are more difficult to observe in the raw tomogram, but inherent variations that reflect the presence of two classes are nevertheless observed.

Furthermore, following the reviewer's suggestion, we now mapped the distribution of the two *T. mirabilis* cartwheel classes onto tomograms, revealing that both classes can occur next to each other within the same centriole (new Figure S8E).

3. Periodicity mismatch

In Fig. 2CD, periodicity of CID has discrepancy from that of the stacked SAS-6 ring (8.5nm and 8.0nm). Do the authors think this is a significant difference or within an error? The same question can occur to other subtomogram averages. It would be nice to show errors as shown in their other manuscript (Fig.3C of Klena et al.) and clarify their idea. If it is systematic difference of periodicity between the stacked ring and CID, this shift will be accumulated through the entire cartwheel region - after 100nm, 8.5nm/8.0nm difference can be accumulated to ~6nm, which should change the entire view of the subtomogram - and the main factor to be classified (periodicity mismatch). This artifact (or influence) should be removed (or separately evaluated) by masking CID (out and in) and run classification separately. By clarifying this, the quality of the major subaverages (mentioned in the previous paragraph) could be improved.

The reviewer wonders whether there might be a periodicity discrepancy within one map, for instance between CID and spokes in the *T. spp.* cartwheel map (Fig. 2C and Fig. 2D). Here, the periodicity determined from the STA maps is 8.5 ± 0.2 nm (SD, N=4) for the CID and 8.0 ± 1.5 nm (SD, N=2) for the spokes. Based on these standard deviations, there is indeed no significant difference between the two, and thus no periodicity discrepancy. The same applies for measurements in *T. agilis* and *T. mirabilis*. The SDs were reported already in the figure legends of the original submission, and we would prefer to leave them there if possible and not mention them in the figures, which are pretty busy as is. We apologize if this was not clear enough in the initial manuscript. Likewise, one may wonder whether there might be periodicity discrepancies between structures from distinct maps, for instance between CID and A-links from *T. spp.* (Fig. 2C and Fig. 3D). Again, the measurements are within error, since the distance between adjacent CIDs is 8.5 ± 0.2 nm (N=4) and between adjacent A-links 8.4 ± 0.4 nm (N=6); a similar conclusion applies for the corresponding measurement comparisons in *T. agilis* and *T. mirabilis*. The figure legends have been altered in the revised manuscript to spell out that there are no significant differences between periodicities (lines 856-858).

Furthermore, we would like to stress that, by definition, STA value are average distances. For instance, in the case of *T. spp.*, the central cartwheel STA was obtained from 511 sub-volumes, and thus the reported N=2 represents the average distance from 511 sub-volumes. Since this is an average, errors can therefore not accumulate over longer distances. This point has also been clarified in the figure legends (line 856-858).

4. Periodicity

They averaged subtomograms extracted with spacing of 252A with initial average as the first template (p.18 Line22). This means they assumed 25nm periodicity from the beginning and excluded different or larger unit size (if they take search range wide, they could detect difference periodicity, but will still be biased by initially assumed 25nm). 25nm average allowed them to see more detail than before (when they

assumed 8nm periodicity), but there is still a risk of bias from references. To avoid this risk, this reviewer would propose classification of randomly extracted (but of course along the cylindrical hub or along the triplet microtubules, so one-dimensionally random picking) subtomograms. This experiment will end up with multiple sub-averages, which are 25nm (or multiple times of that) shifted from each other. Then it will prove their assumption.

We agree with the reviewer that in theory the choice of periodicity could introduce a bias. This is why we have chosen a larger step size than in our initial work, corresponding to ~3x the major periodicity of ~8.5 nm observed in the power spectrum of the sub-volumes, as mentioned above. Regardless, following the reviewer's suggestion, we have now explored other types of periodicities by re-analyzing the dataset through extraction of non-overlapping sub-volumes along the proximal-distal centriole axis. In doing so, we randomized the starting position of the first box between tomograms, reaching the same goal as with random picking but maximizing the number of sub-volumes. We carried out this analysis for all *T. spp.*, *T. agilis* and *T. mirabilis* cartwheel classes, and found no notable differences that would affect the conclusions of the manuscript compared to the initial overlapping sub-volume classification, albeit generally with a noisier STA due to the lower number of sub-volumes. A comparison of the two approaches is provided in Rebuttal Figure 2. Moreover, all the points regarding the choice of periodicity have been further clarified in the expanded Materials and Methods section (pages 19-21).

Rebuttal Figure 2: Reclassification with non-overlapping sub-volumes

(A-F) Transverse (top) and longitudinal (bottom) views of *T. spp.* (A, B) *T. agilis* (C, D) and *T. mirabilis* (E, F) central cartwheel 3D maps. The final maps reported in the manuscript (A, C, E) were generated with a 25 nm step size, yielding overlapping sub-volumes, whereas the maps in (B, D, F) were generated from non-overlapping sub-volumes, with no notable differences between the two that would affect the conclusions of the manuscript.

Minor points:

They discussed difference of stacked SAS-6 rings in the cartwheel from various species. How much is the sequence difference of SAS-6 among these species?

Unfortunately, no genomic or transcriptomic data has been published for the species investigated here, although the sparse molecular data available from small subunit rRNA sequences allows one to establish an overall molecular phylogeny. We previously identified a SAS-6 homologue in *T. agilis* (Guichard et al. 2013), which shares 20 % identity and 45 % similarity with *C. reinhardtii* SAS-6. Despite low sequence conservation, the structural conservation of SAS-6 is predicted to be high between the two organisms (Guichard et al. 2013). We apologize if these points were

not expressed sufficiently clearly in the initial rendition and have adapted the wording in the revised manuscript (lines 325-332).

| *Are the authors sure that CID is nine-fold symmetric? It is not trivial.*

We thank the reviewer for bringing up this interesting point. We have applied 9-fold symmetrization to the entire central cartwheel comprising spokes, hub and CID/ fCID, a choice guided by the apparent 9-fold symmetry of the spokes and peripheral element. We investigated the impact of symmetrization on the CID by relaxing symmetry from C9 to C1 during refinement, but did not observe a difference, and thus continued with C9 symmetry, which improves map resolution by S/N ratio enhancement and additional missing wedge compensation. In addition, we have also analyzed the CID without symmetrization, as reported in Figure S7 (previously: Fig. S6). Note that these maps were generated with larger sub-volumes centered on the spokes to comprise hub, spokes and microtubule triplets, explaining the resulting lower resolution, as the missing wedge is not compensated. Despite these limitations, however, the unsymmetrized CID shown in Figure S7A and S7E resembles the one in the symmetrized maps of Figure 2, indicating that the CID indeed exhibits 9-fold radial symmetry. That this is the case is spelled out explicitly in the revised manuscript (lines 1145-1147).

| *p.7 Line21 "Fig.S1D-O": D-L*

We apologize for this mislabeling, which has been corrected.

| *p.8 Line1: It would be nice if more detailed description about MIPs, correlating to recent high resolution works from Bui and Brown labs.*

We agree that a more detailed comparison with recently identified axonemal MIPs was needed and have expanded the discussion accordingly (lines 481-492).

| *p.9 Line6 "Focused 3D classification...": This sentence is unclear.*

Apologies about this. The original statement might have been unclear because of the word "respectively" that was used incorrectly. The sentence has been clarified (lines 187-189).

| *p.18 5 lines from bottom "S6C, S6F": How can these panels be power spectra to measure spacing? Typo?*

We thank the reviewer for having spotted this typo, which has been corrected (now Fig. S4C and S4F).

| *Fig.1C: Another cross-section from the distal region will be helpful. A longer scale bar is better for readers' understanding.*

We understand that the reviewer is curious about the distal region, and cross-section views of resin-embedded sections from *T. agilis* are available and could be provided if necessary. However, given that the focus of the manuscript is strictly on the cartwheel-bearing proximal region, we felt that featuring the distal region in detail would break the narrative. Therefore, we suggest to keep Figure 1 as in the original manuscript. Following the reviewer's suggestion, we increased the size of the scale bars from 10 nm to 20 nm in Figure 1C as well as in the corresponding Figure S8C.

| *p.29 Line6: pin -> pink*

The typo was corrected.

| *Fig.S6F: It would be informative if the subclasses (25% and 20%) are distinguished in this mapping.*

As per the reviewer's request, we provide in Rebuttal Figure 3 a side-by-side comparison of the *T. agilis* 25 % and 20 % classes centered on the spokes, which are noisier than the composite 45 % class due to the lower number of sub-volumes in each sub-class. Given that there are no notable differences between the two maps that would affect any of the conclusions of the manuscript, we feel it is best to keep what is now Figure S7F (previously: Fig. S6F) unchanged in the revised manuscript.

Rebuttal Figure 3: Polar centriolar cartwheel upon sub-classification

(A-C) 3D transverse views of non-symmetrized STA centered on the spokes to jointly show the central cartwheel and peripheral elements in the *T. agilis* 45 % class (A), as well as separately in the 25 % class (B) and 20% class (C). No notable differences are apparent following such re-classification, apart from the output being noisier due to the lower number of sub-volumes in each sub-class.

A figure to explain the classification scheme will help readers understand. How many subtomograms did classification started? Were the 45% class classified into two (25% and 20%) groups by two-step classification or at once (the entire subtomograms were classified into three groups directly)?

We thank the reviewer for this useful suggestion. As a result, we have generated a new Supplemental Figure S1G-J that provides a graphical overview of the classification scheme, together with sub-volume numbers for all deposited maps, thus nicely complementing Table S1.

Reviewer #1 (Significance (Required)):

Nevertheless, this work demonstrated capability of cellular cryo-ET, especially analysis of structural heterogeneity. Thus, while biological topics handled are rather specialized for cilia from flagellate, this work will attract attention of any biologist interested in molecular structure in vivo. It is worth for publication in a high journal after addressing the points above. This reviewer believes that the authors can address these points easily with additional analysis.

We reiterate our thanks to this reviewer for her/his favorable evaluation and detailed suggestions, which enabled us to generate a strengthened manuscript.

Reviewer #2 (Evidence, reproducibility and clarity (Required)):

Here, Nazarov and colleagues report sub-tomogram average (STA) maps of centrioles with 16 to 40 Å resolution from *Trichonympha* spp., *Trichonympha agilis*, and *Teranympha mirabilis*. Even though the authors have previously described the centriole architecture of *T. spp*, these STA maps of higher resolution revealed new features of centrioles, like polarized Cartwheel Inner Density (CID) and the pinhead. They also observed Filament-like structure (FLS) from *T. mirabilis* which seems to correspond to the CID from other species. Interestingly, they suggest that one and two SASS6 rings are stacked in an alternative fashion to make the central hub in *T. mirabilis* (Figure 5). The following issue should be addressed:

Major points

1. Figure 4E. Authors mentioned in the manuscript that "We observed that every other double hub units in the 36% *T. mirabilis* class appears to exhibit a slight tilt angle relative to the vertical axis". When I see the other side, it does not seem to be tilted. Could the authors explain this?

We apologize that this aspect was not explained in sufficient detail. The left and right sides of the hub indeed appeared different in transverse views across the cartwheel center (previous Fig. 4E). This was because the area we selected in the original submission was centered on one emanating spoke. Due to the 9-fold symmetry one spoke density was selected on the right side, while the region between two spokes was displayed on the left side (as was illustrated by the slice across the center in previous Figure 4A; dashed rectangles in 4.0 nm panel). We have now selected a larger area to include spokes from both sides of the hub and thus better visualize this offset as shown in the modified Figure 4D-E.

Minor Points

1. Page 11, I think Fig. 9G indicates Fig. S9G.

We apologize for this typo that has been corrected.

Reviewer #2 (Significance (Required)):

I believe these results are of interest for all centrosome researchers and would like to recommend this manuscript be published in the EMBO journal which is affiliated with the Review Commons.

We thank the reviewer for the recommendation to submit the revised manuscript to *EMBO Journal*, which we have followed.

Reviewer #3 (Evidence, reproducibility and clarity (Required)):

*In this manuscript Nazrov et al., use cryo-electron tomography (CET) to analyse the structure of the centriole cartwheel. The Gonczy lab have previously generated a ground-breaking structure of the cartwheel from *Trichonympha* spp (*T. spp.*) (Guichard et al., *Science*, 2012; Guichard et al., *Curr. Biol.*, 2013). This work is a direct continuation of those studies but using modern technology to get higher resolution images of the *T. spp.* cartwheel and comparing this to the cartwheel from *Trichonympha agilis* and from another distantly related flagellate *Teranympha mirabilis*.*

*The data is generally well presented and of high quality. I am not an expert in CET, so it would be advisable to get the opinion from a reviewer who is, but the Gonczy lab are experienced in these techniques so I would not anticipate any problems. I have to admit that the title of the paper did not excite me, and I expected this to be a very worthy, but incremental study. It was a pleasure to find out that the extra detail provided by the increased resolution has revealed several new and unexpected features that have important implications for our understanding of cartwheel assembly and function. Most important are the potential asymmetry of the cartwheel hub, apparent variations in the packing mechanism of the stacked rings (even within the same cartwheel), and the potential offsetting of ring stacking. These findings will be of great interest to the field, and so I am strongly supportive of publication in *The EMBO Journal*. I have only a few points that I think the authors should consider.*

We thank the reviewer for this positive feedback and the recommendation to submit to *EMBO Journal*, which we hereby follow.

Prompted by the comment of the reviewer, we revised the title to make it more informative and appealing to readers: "Novel features of centriole polarity and cartwheel stacking revealed by cryo-tomography".

1. Nazarov et al., conclude that the cartwheel structure is intrinsically asymmetric. This is most convincingly based on the displacement of the CID within the hub, but they state that the Discussion that the potential offset between the Sas-6 double rings generates an inherently polar structure. I didn't understand why this is the case. Looking at Fig. S9A,B I can see that the offset in B could tilt to the left (as shown here) or to the right (if the structure was flipped by 180°). But I couldn't see how this makes this structure polar in the sense that a molecule coming into dock with the structure could only bind to one side of the offset structure shown in B, but to both sides of the aligned structure shown in A. I think this needs to be explained better, as it is crucial to understand where any potential polarity in the cartwheel structure comes from.

We apologize for not having been sufficiently clear about how two SAS-6 rings with an offset could impart organelle polarity. The reviewer is correct that an offset between superimposed rings alone is not sufficient to generate polarity at a larger scale. The important point we would like to stress, however, is that we discovered concerted polarity in multiple locations, from the central hub to the peripheral elements as illustrated in Fig. S7C-D, S7G-H, S7K-L and S7O-P (previously: Fig. S6). Prompted by the reviewer's comment, we now better emphasize the asymmetric tilt angles of merging spokes, as highlighted also in the improved Figure S7. This asymmetric spoke tilt angle allows one to discriminate the proximal and distal side of a double SAS-6 ring, which is now explained better in the text (lines 259-263 & 502-510).

2. Related to this last point, in a co-submitted paper Klena et al. do not report such an asymmetry in the hub structures they have solved from several different species (neither in the tilting of the hub, or the displacement of the CID). I think it would be worth both sets of authors commenting on this point.

We agree that comparing and contrasting the results of the two companion manuscripts is important and we have updated the text as a consequence in several places (lines 444, 467, 507, 536, 985, 1000). We know from our previous work (Guichard et al. 2013) that the asymmetry of the hub and spoke is not visible at lower resolution. In the accompanying manuscript by Klena et al., no offset in the hub or asymmetric CID localization is reported, probably due to lower resolution and differences between species.

3. The authors data strongly suggests that the *T. ag.* and *Te. mir.* hubs are composed of a mixture of single and double Sas-6 rings. In contrast, the *T. spp. cartwheel* only has a single class of rings, but it wasn't absolutely clear if the authors think this comprises a single or double ring. In the text it is presented as though the elongation of the hub densities in the vertical direction is a new feature of the *T. ag cartwheel* (Fig.2H,I), but to me it looks as though this is also apparent in the *T. spp. cartwheel* (Fig.2C,D). The authors should address this directly and, if they believe that *T. spp.* has a double ring, they should comment on whether this more regular structure seems to have offset rings. If not, then the offset rings are unlikely to be the source of asymmetry that leads to the asymmetric displacement of the CID. Finally, if the authors think these are double rings, they should also be clear that they would now slightly re-interpret their original *T. spp. cartwheel* model (Figure 2, Guichard et al., *Curr. Biol.*). There is no embarrassment in this—a higher resolution structure has simply revealed more detail.

We apologize if the conclusions drawn about *T. spp. cartwheel* hubs were not sufficiently clearly expressed. Like the reviewer, we think that elongated hub elements are also discernible in *T. spp.*, something that is also illustrated by the intensity plot profile in Figure 2C (double peaks on light blue line). These points are spelled out more explicitly in the revised manuscript (lines 177-179). In addition, to emphasize the conservation of the double hub units in both *Trichonympha* species, we have likewise adapted the text for *T. agilis* (lines 198-201).

As for the offset observed within *T. spp.* spoke densities in Figure S10H, we interpret this as evidence for an offset of the double ring at the level of the hub, although we have not observed such offset in *T. spp.* for reasons that are unclear. The fact that this revises our previous interpretation based on a lower resolution map of *T. spp.* was already mentioned in the initial submission but is now better emphasized (lines 171-172 & 179-181).

4. The authors conclude that *T. mirabilis* cartwheels lack a CID and instead have a filament-like structure (FLS). I wonder whether it is more likely that the FLS is really a highly derived CID that appears to be structurally distinct when analysed in this way, but that will ultimately have a similar molecular composition. This situation might be analogous to the central tube in *C. elegans*, which by EM appears to be distinct from the central cartwheel seen in most other species, but is of course still composed of Sas-6. This historical tube/cartwheel nomenclature is now cumbersome to deal with, so perhaps it would be better to be cautious and not give the *T. mirabilis* structure a completely new name—how about "unusual CID" (uCID).

We share the view that the CID and the "FLS" –the term used in the initial submission– may have a related molecular composition and function, as we had also speculated in the discussion of the original submission. Following the reviewer's suggestion, and in

an effort to have a more uniform nomenclature, we propose to dub the *T. mirabilis* structure “filamentous CID” (fCID). This highlights better the similar location of these two entities and their potential shared function, while stressing the filamentous nature of the fCID. We further emphasize this point by providing the new Figure 6A to compare the presence of the two entities in select species. The discussion has also been adapted accordingly (pages 13-14).

| *Reviewer #3 (Significance (Required))*: see above

Prof. Pierre Gönczy
Swiss Federal Institute of Technology (EPFL)
Swiss Institute for Experimental Cancer Research (ISREC), School of Life Sciences
Station 19
Lausanne CH-1015
Switzerland

31st Jul 2020

Re: EMBOJ-2020-106249
Novel features of centriole polarity and cartwheel stacking revealed by cryo-tomography

Thank you again for having transferred your already revised Review Commons manuscript to The EMBO Journal for our consideration. In light of the positive previous assessment and the interest of the subject of the study, I decided to treat it essentially as a revision, sending it back to referee 1 for assessing your responses to the various technical concerns originally raised by this reviewers. As you will see from the comments copied below, referee 1 is fully satisfied with your revision, and we shall therefore be happy to accept this work for The EMBO Journal, pending incorporation of various editorial points as detailed below:

I am therefore returning the manuscript to you for a final round of minor revision, to allow you to make these adjustments and upload all modified files. Once we will have received them, we should be ready to proceed with formal acceptance and production of the manuscript.

With best regards,

Hartmut

Hartmut Vodermaier, PhD
Senior Editor / The EMBO Journal
h.vodermaier@embojournal.org

REFEREE 1:

Nazarov and his colleagues addressed all the points from this reviewer. Especially they conducted additional analysis and (1) exploited currently possible classification schemes (ended up with their original result) and (2) demonstrated 25nm periodicity by analyzing randomly (thus non-biased way) extracted subtomograms. This reviewer appreciates their praisable revision work and strongly recommend publication of this manuscript in the EMBO Journal.

Swiss Federal Institute of
Technology Lausanne

Pierre Gönczy

Swiss Institute for Experimental
Cancer Research (ISREC)

School of Life Sciences

Swiss Federal Institute of
Technology Lausanne (EPFL)

pierre.gonczy@epfl.ch

Phone: +41 21 693 07 11

<https://gonczy-lab.epfl.ch/>

Lausanne, August 11th, 2020

Re: EMBOJ-2020-106249

Thank you for your message dated July 31st, 2020, informing us that you shall be happy to accept our work for publication in *The EMBO Journal*.

As per your request, we have attended to the following editorial points:

- completed author checklist
- provided legends for Figures 2 and 3, as well as what used to be Figures S2-S4, with the flow of panels being ordered alphabetically
- reorganized the data display in accordance with your author guide (whereby S6 > EV1, S7 > EV2, S8 > EV3, S9 > EV4 and S10 > EV5, with the other Supplementary figures now incorporated into the Appendix, together with the Supplementary Tables)
- provided a two sentence summary, together with bullet points, as well as an accompanying schematic summarizing our findings

We trust that with these changes, you will find our work ready for publication in *The EMBO Journal*.

Thank you for submitting your final revised manuscript for our consideration. I am pleased to inform you that we have now accepted it for publication in the EMBO Journal!

Corresponding Author Name: Pierre Gönczy

Manuscript Number: EMBOJ-2020-106249